# CAUSAL ANALYSIS OF SOCIAL BIAS TOWARDS FACES IN CLIP

## ABSTRACT

We propose the first experimental study to causally measure bias in social perception in the latent space of multi-modal models. Previous studies compute *correlations* between a model's social judgments and protected attributes, such as race, age, and gender, using *observational* wild-collected human-annotated datasets, such as FairFace. In order to establish *causal* links between protected attributes and algorithmic bias, we use a synthetic dataset of face images instead, Causal-Face, where both legally protected attributes and potential confound attributes, such as facial expression, lighting, and pose, are controlled independently and systematically, and thus allow an *experimental* exploration, which lets us reach causal conclusions. Our analysis is based on measuring cosine similarities between images and word prompts, including valence words drawn from the two leading social psychology theories elucidating human stereotypes: The ABC Model and the Stereotype Content Model. We find that non-protected attributes are powerful confounds and profoundly influence social perception, injecting variability in measurements whose size is comparable to that induced by legally protected attributes. Clear intersecting biases of race, gender, and age only emerge when these unprotected attributes are controlled for, which is only possible using CausalFace. Real-world datasets do not permit a similar level of insight due to spurious correlations introduced by uncontrolled attributes and a lack of specific annotations.

## 1 INTRODUCTION

Automated intelligent systems are increasingly being applied in manufacturing, entertainment, transportation, health, security, safety, and education. Such "artificial intelligence" (AI) systems can analyze and synthesize images, sounds, speech, and language and are trained on large corpora of data and through interaction with humans, using machine learning. Whenever automated systems interact with humans or make decisions that can affect the health and welfare of humans, it is important to ensure that the dignity and welfare of users and stakeholders is preserved. An even-handed treatment of every person is one crucial concern. The starting point towards this goal is detecting and measuring potential performance biases vis-a-vis protected attributes such as age, gender, and race (Kearns & Roth, 2019).

A crucial aspect of measuring bias is identifying the *cause* of it. This is for two reasons. First, social justice demands that if the cause of bias is a protected attribute, e.g., being female causes a higher rejection rate in loan applications, the bias is addressed as soon as possible. Second, only by knowing the cause of bias can engineers address and fix it. It is important to remember that *correlation does not imply causation*, and establishing causes is more difficult than finding correlations. *Experiments*, i.e., the systematic modification of one variable at a time, were developed precisely for this purpose.

We focus on the question of how to measure large multimodal model bias in the social judgments of human faces. We chose CLIP (Radford et al., 2021) as the representative to be tested in our experiments, it is the state-of-the-art vision-language model widely used for zero-shot classification (Radford et al., 2021), image retrieval (Agarwal et al., 2021), or for guiding generative text-to-image models (Rombach et al., 2022; Ramesh et al., 2022; Saharia et al., 2022; Balaji et al., 2022). CLIP models are trained on large text-image-pair datasets (e.g., Schuhmann et al. (2021)) gathered from

various internet sources. Naturally, image captions may contain human social biases, some of which may trickle into the model (Steed & Caliskan, 2021).

To quantify social judgment, we make use of two established theories from social psychology—the Stereotype Content Model (Fiske et al., 2007) and the ABC-Model (Koch et al., 2016)—which provide well-established frameworks for measuring stereotypes held by humans.

Current literature is mostly correlative. Bias is measured using test datasets that are collected in natural settings and annotated for legally protected attributes like race and gender. Algorithmic performance is measured as a function of such annotations. Unfortunately, the distribution of non-protected attributes—such as (in the case of face images) lighting, pose, and facial expression or simply image color statistics (Meister et al., 2023)—across these groups is inevitably correlated with the protected attributes. Thus, when bias is found, it is unclear whether bias belongs to the algorithm, test data, or both. Additionally, it is difficult to obtain sufficiently diverse samples across all intersectional groups. Thus, the prevalent approach only allows for correlational conclusions and often does not achieve statistical significance across all intersectional groups in the dataset.

To address these issues, we propose to adopt an *experimental* approach by employing a synthetic dataset of face images that are generated using generative adversarial networks (GANs) (Liang et al., 2023), where a number of protected attributes (race, gender, age) and additional non-protected attributes (pose, expression, and lighting) are manipulated systematically and independently of each other. This dataset allows us to directly examine the impact of each attribute, rather than relying on correlations, and thus arrive at causal imputations of bias.

It is imperative to emphasize that using discrete categories such as race, gender, or age is a methodological approximation for evaluating biases within models. Recognizing the richness and complexity of human identities, we advise against using such categories to pigeonhole or label individuals in real-world contexts. We adopt a simplified race categorization, limiting ourselves to Asian, Black, and White. As will be clear later, when we refer to "race," we refer to social constructs, as in the perception of human observers, rather than biological realities.

## 2 RELATED WORK

### 2.1 RESEARCHING BIAS IN CLIP

Fairness in CLIP has been explored by a number of authors, some reporting racial bias in classification (Agarwal et al., 2021), gender bias in neutral text (Dehouche, 2021), language-specific biases (Wang et al., 2021), and learned stereotypes (Wolfe et al., 2022). Such biases can significantly impact image retrieval, potentially leading to unequal treatment (Geyik et al., 2019).

Bias of CLIP-guided generative text-to-image models has also been studied (Orgad et al., 2023; Zhang et al., 2023; Cho et al., 2022) by systematically prompting the model and analyzing the generated image output (Luccioni et al., 2023)—for instance, associating the image of a CEO more frequently with males than females (OpenAi, 2022; Bianchi et al., 2023). The prompting strategy in these studies is not anchored by a specific theoretical framework.

We follow another approach, namely estimating CLIP bias via image retrieval. More in detail, we start from an image dataset, annotated by attributes of interest and a set of text prompts. We iteratively consider one image and a text prompt, infer their CLIP embeddings, and compute their cosine similarity within their shared latent space (Radford et al., 2021). Investigating CLIP embeddings directly provides distinct advantages over solely analyzing the output of specific prompts, particularly when exploring intersectionality. Intersectionality recognizes that an individual's identity and experiences are not singular but are informed by multiple, interconnected layers of discrimination and disadvantage, including racism, sexism, and classism.

Social categorization and stereotyping are fundamental social-cognitive processes (Fiske & Neuberg, 1990). Social Psychology has systematized those processes along the dimensions of Warmth and Competence (Fiske et al., 2007) . A more recent model has subdivided and renamed those dimensions (Koch et al., 2016). We draw from this theoretical foundation to construct text prompts in our experiments. A few authors in the representation learning literature have seen value in this approach (Cao & Kosinski, 2023; Ungless et al., 2022; Fraser et al., 2023). Otterbacher et al. (2017)

found that biases somewhat align with social perceptions of warmth and competence. Unlike Cao & Kosinski (2023); Ungless et al. (2022), our study shifts the lens from unimodal (language) to multimodal (text+image) models. Furthermore, unlike Fraser et al. (2023), we directly work with embedding space rather than analyzing generation outputs.

Finally, and crucially, CLIP bias research relies on observational data, while only a few have explored utilizing synthetic images: Wolfe et al. (2022) use a GAN to "morph" face photographs to measure CLIP's association of multiracial people to minority classes. This method provides limited control over image attributes and is, therefore, not suitable for our analysis.

## 2.2 STEREOTYPE MODELS IN SOCIAL PSYCHOLOGY

Given the reflection of human-like biases in Generative AI, our study employs two leading theoretical frameworks from social psychology that have long been recognized for delineating the primary dimensions of beliefs about stereotypes.

The *Stereotype Content Model* (Fiske et al., 2007) (SCM) measures social perceptions in a two-dimensional space of warmth (W) and competence (C) based on extensive evidence that the two-variable warmth-competence reduction robustly explains a surprising amount of variation across perceptions and behavioral reactions to social categories. "Warmth" quantifies the perception of how good or bad another person's intentions are. "Competence" refers to the perception of a person's capability of acting on their intentions (Fiske et al., 2007). Both categories are described by six adjectives (Table A.2) each.

A more recent framework, the *ABC Model* (Koch et al., 2016), proposes beliefs as an alternative central dimension and subdivides the Warmth and Competence dimensions differently into two categories, specifically: "Agency" or socio-economic success, conservative-progressive "Beliefs", and a group's "Communion" or warmth. These dimensions are divided into positive and negative valence dimensions containing four to six adjectives each (Table A.2). We denote these dimensions as A+, A-, B+, B-, C+, and C-.

## 3 METHODS

### 3.1 MEASURING BIAS

We assess bias in multimodal models by computing the cosine similarities between text embeddings and image embeddings, following CLIP's image retrieval metric (Radford et al., 2021). In our case, we calculate the similarity between a set of images $I$ (e.g., all face images of a demographic group) with a dimension $D$ (see Section 2.2) as

$$\cos(\boldsymbol{I}, \boldsymbol{D}) = \mathrm{mean}_{\boldsymbol{i} \in \boldsymbol{I}, \boldsymbol{d} \in \boldsymbol{D}} \left( \mathrm{mean}_{t \in T} \cos(\boldsymbol{i}, t(\boldsymbol{d})) \right), \tag{1}$$

where $T$ is a set of contextualized prompt templates (e.g., "a photo of a <adjective> person", see Table A.1 for all templates). Averaging similarities over a set of templates is a common practice to make results more robust (Berg et al., 2022).

The Word Embedding Association Test (*WEAT*) was designed to measure the differential association between two sets of two target dimensions and two sets of polar attributes in text embeddings (Caliskan et al., 2017). The WEAT has been adapted to discern bias within image embeddings (Steed & Caliskan, 2021) (where image embeddings represent attributes) and was validated through human evaluations (Wolfe et al., 2022). We utilize the Single-Category Word Embedding Association Test (SC-WEAT, Appendix A.2).

Bias can also be quantified through *Markedness*. Given that CLIP deciphers unstructured visuals via linguistic patterns, categorizations could be community-specific. Linguistic structures may harbor biases from dominant groups, leading to "unmarked" (default) or "marked" (non-default) categorizations. Sociologically, markedness amplifies the disparities between marginalized and dominant entities. We aim to discern if CLIP mirrors such linguistic biases in its visual interpretations (Wolfe & Caliskan, 2022). Markedness as a metric describes the fraction of images containing a specific attribute, where the similarity with a neutral text prompt is higher than that with a text prompt containing the said attributes (Appendix A.3).

## 3.2 IMAGE DATASETS

The FairFace dataset (Karkkainen & Joo, 2021) is a real-world image dataset commonly used in studies on AI fairness and bias. It comprises 108,501 face images labeled with information about race, gender, and age. It offers a balanced representation across seven racial categories. We use only the overlapping racial categories of all three datasets, which are Asian, White, and Black, and images that are annotated with an age of 20 years and older. Therefore, we generate a subset of 38,744 images from the Fairface training subset (Figure D.1).

Similar to FairFace, the UTKFace dataset (Zhang et al., 2017) face images, each annotated with demographic details such as race, gender, and age. From its 23,708 face images, we sampled 14,630 images using the same criteria as for Fairface (Figure D.2).

CausalFace[1] is a synthetic face dataset introduced by Liang et al. (2023). Utilizing Generative Adversarial Networks (GANs), the authors create numerous "pseudo-identities" defined by a seed. An identity can be represented by three different races (Asian, Black, and White) and as male or female. Importantly, all these prototypes are forced to be as similar as possible in all other attributes except their race and gender (e.g., facial proportions, clothing, background). In addition, each prototype image was systematically modified along four semantic attributes (pose, age, expression, and lighting). Example images of the data set can be found in Appendix D.2. In this work, we use images from 100 different seeds, each containing six race-gender combinations. In addition, we sample variations for age (9), smiling (9), pose (4), and lighting (7). In total, there are 18,000 unique images included. The unique property of this dataset is that it changes image attributes in an isolated manner, allowing for causal interpretation. To support this, Liang et al. (2023) visually inspected whether manipulating one attribute does not induce unwanted changes in other attributes. Additionally, human annotators confirmed that facial attribute manipulations are equally effective across demographic groups (Liang et al., 2023, Fig. 3). A nuanced view on the causal interpretation of the dataset can be found in the Section 5.

## 3.3 COMPARING INTRODUCED VARIATION

CausalFace uniquely enables comparison of variations due to legally protected attributes (e.g., age, race, gender) against those from confounds (e.g., smiling, lighting, pose). Given the varying number of levels within each category (for instance, "smiling" has ten levels, whereas "gender" is binary, consisting of only male and female), the standard deviation is not universally applicable. To address this, we use a sampling strategy. We sample two values for $x_1, x_2 \sim X$ for one specific CausalFace dimension and select two image embeddings $i_1(x = x_1), i_2(x = x_2)$ where the value for all other dimensions are equal. Then the difference in cosine similarities between the two selected images and a text embedding $t$ is defined by: $\Delta(t, i_1, i_2) = |\cos(i_1, t) - \cos(i_2, t)|$. This process is repeated 1,000 times, leading us to analyze the distribution of these $\Delta$ values. For ordinal attributes such as age and smiling, we introduce additional constraints. To ensure a perceptually significant change in image appearance, we mandate that two samples must differ by at least a threshold: set at 0.7 for age and 1.1 for smiling[2]. These thresholds ensure that the images being compared are visually distinct in the context of the attribute under consideration.

## 4 RESULTS

### 4.1 CAUSALFACE AND REAL-WORLD DATASETS SHOW SIMILAR BIAS METRICS

Since CausalFace contains artificially generated images, one could question the dataset's representativeness, as these faces might have different properties w.r.t. bias compared to real photos. To address the representativeness of our synthetic data, we first treat CausalFace as a statistical dataset, omitting the additional information about non-sensitive attributes and possibly within-seed analyses. In the following, we investigate three metrics—mean cosine similarities, markedness, and the WEAT, to find out whether CausalFace behaves similarly in comparison to two observational datasets: FairFace and UTKFace.

---

[1]The term "CausalFace" was introduced by us, as the authors do not name their dataset.

[2]The impact of the thresholds can be better understood by studying the scale of smiling and age, depicted in Figure D.3 and Figure D.4.

Table 1: **CausalFace, FairFace, and UTKFace have similar overall statistics in CLIP latent space.** Markedness indicates the preference (in %) for a neutral prompt over a race or gender-specific one. White is mostly unmarked while all other categories are marked. The mean cosine similarities (in %) represent averages of cosine similarities from both the SCM and the positive dimensions of the ABC models.

| Image Category | Markedness | | | Mean Cosine Similarity | | |
|---|---|---|---|---|---|---|
| | CausalFace | FairFace | UTKFace | CausalFace | FairFace | UTKFace |
| White | 46.00 | 54.12 | 32.58 | 23.29 | 23.07 | 22.25 |
| Black | 0.73 | 3.92 | 2.89 | 22.94 | 22.61 | 21.93 |
| Asian | 0.05 | 3.88 | 4.07 | 23.78 | 23.42 | 22.68 |
| male | 3.06 | 0.00 | 20.84 | 23.22 | 23.07 | 22.03 |
| female | 6.58 | 0.00 | 11.59 | 23.45 | 23.17 | 22.45 |

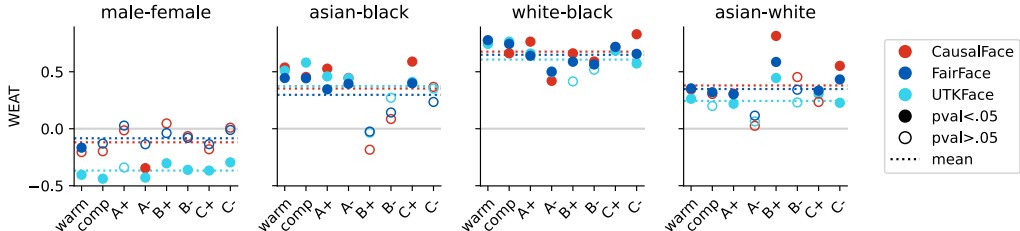

Figure 1: **WEAT scores in CausalFace are comparable to those in FairFace and UTKFace.** The dots represent the mean difference per dimensions (e.g., A+). The grey horizontal line marks the zero value. The different colors indicate whether data comes from CausalFace, FairFace or UTKFace. Shapes indicate whether the permutation test indicated a $p$-value $< .05$ or not.

*Mean cosine similarities* are calculated as the mean of cosine similarities between an image category and all positive items of the social perception dimensions ( Equation (1)). The mean cosine similarities exhibit uniform trends across both datasets. Specifically, Asians consistently display the higher cosine similarities than Whites, who exhibit higher similarities than Blacks (Table 1).

*Markedness* is the relative preference frequency for a neutral prompt over a race or gender-specific one (Wolfe & Caliskan, 2022) (Section 3.1). In all three datasets, CLIP shows a preference for an unspecified prompt over one that specifies the race as "White." The percentages vary, ranging from 32.58%, 46.00%, to 54.12%. CausalFace falls in the middle, exhibiting neither the highest nor the lowest preference for an unspecified term. CausalFace and FairFace both display a significant decrease in preferring an unmarked prompt for the remaining four race or gender categories. This decline is more pronounced in FairFace than in CausalFace. In contrast, UTKFace shows a less marked decrease. (Table 1).

The visualizations of WEAT scores further underscore the alignment between CausalFace and the two observational datasets (Figure 1). Dotted lines, each in a unique color, indicate the average scores across various social perception dimensions for each dataset. The most striking similarities in means are observed when comparing Asians to Blacks and Whites to Blacks. In the cases of Asian versus White and Male versus Female comparisons, CausalFace and FairFace demonstrate closely matched scores, while UTKFace records lower WEAT values in both scenarios. Notably, across all five comparisons, the variance in mean scores within a single panel is less pronounced than the differences observed between panels.

By evaluating mean cosine similarities, markedness, and WEAT scores, we observe that CausalFace displays trends that closely mirror those in FairFace and UTKFace. Following Geyik et al. (2019) , we have also examined additional metrics such as Skew@k, MaxSkew@k which present a consistent narrative (Appendix B.1). Consequently, we deduce that the synthetic image dataset offers a valuable foundation for examining biases present in real-world images.

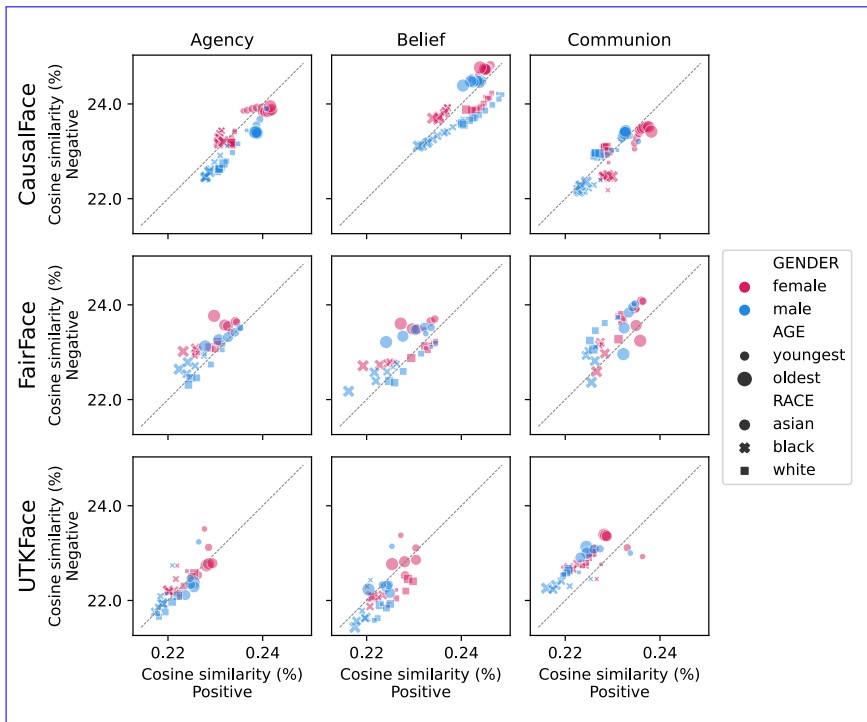

Figure 2: **Intersectional groups cluster in CausalFace. Not so in FairFace and UTKFace**. Comparison of social perceptions by intersectional group: Markers depict the means of social perceptions for different intersectional groups. The x-axis represents positive, and the y-axis negative valence. One can observe similar cluster locations w.r.t. race and gender in all data sets. However, the effect of age is more clearly visible in CausalFace due to successful noise reduction.

### 4.2 CAUSALFACE REVEALS CLEAR INTERSECTIONAL PATTERNS

Having examined bias at an aggregate level, focusing primarily on the commonly studied legally protected attributes of race and gender, we now turn our attention to a more detailed investigation of the intersecting dimensions of gender, age, and race.

Comparing positive and negative valence items, highlights differences between CausalFace and the observational datasets of FairFace and UTKFace (Figure 2). While CausalFace showcases distinct clusters for combinations of race, gender, and age, neither FairFace nor UTKFace display distinct clusters. The clarity in the CausalFace representation can most likely be attributed to the controlled environment where variables like lighting and pose are held constant. FairFace and UTKFace, on the other hand, do not offer this level of control, making their representation more susceptible to external factors. As a result, the synthetic nature of CausalFace offers a unique lens, illuminating intersecting biases that might remain obscured within a purely correlational dataset.

A striking observation in all subplots is the notable positive correlation. This suggests that if a group registers high scores on the positive dimension, it simultaneously reflects high scores on the negative dimension. This is counterintuitive, as one would expect these opposing dimensions to correlate negatively. This observation can be explained as text embeddings of related dimensions with opposing valence are grouped closer in CLIP's embedding space than unrelated dimensions (see Appendix C for a detailed analysis on this topic). In the context of our bias analysis, it is important to highlight the pivotal distinction between valence and intensity: We use "valence" as the direction of change in cosine similarity (increasing or decreasing) and "intensity" as the magnitude or absolute difference in change. While differences in valence are negligible, variations in intensity for various intersecting groups are salient. Intriguingly, across all three dimensions, Asians exhibit the highest intensity. Blacks, in contrast, register the lowest, with Whites occupying an intermediate position. This is true for all three datasets and aligns results of mean cosine similarities (Table 1).

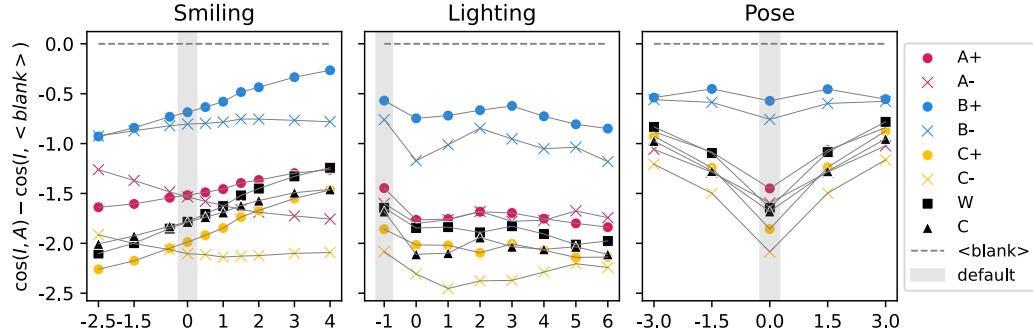

(a) **Smiling and pose systematically influence social perception.** The three panels depict the variations in smiling, lighting, and pose on the x-axis. The cosine similarity of a neutral text prompt is depicted by the grey dashed line, serving as a reference. Deviations from this reference, caused by adding a social perception item to text prompts, are represented as dots connected by thin grey lines. While both lighting and pose exhibit structured patterns, the effects of lighting appear more stochastic in nature. Illustration showing actual cosine similarities on the y-scale is shown in Figure B.2.

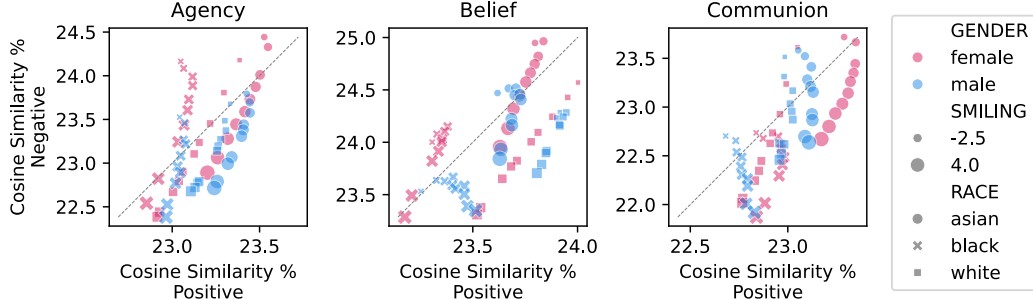

(b) **Facial expression has a differential impact by intersecting groups.** Smiling causes a more pronounced change in social perceptions of women than men. Black males show a strikingly different pattern for Belief.

Figure 3: **Non-protected attributes structurally impact social perception.**

The age-related effects observed in CausalFace across the belief dimension are especially noteworthy. Here, the line formed by the cross (Black) and square (White) markers is more extended than the one outlined by the round markers (Asian). This suggests a more pronounced age effect for Black and White groups than Asians within the belief dimension. "Agency" presents a contrasting age effect for male and female Asians: as Asian males age, intensity decreases. Conversely, older Asian females demonstrate heightened positive scores on the agency dimension, while the negative agency dimension remains neutral.

FairFace and UTKFace, by contrast, are considerably more variable. Although, on average, Asians might score the highest in intensity, the data is muddled, with factors like age playing a significant role. For instance, some younger Asians register as low in intensity as certain older black individuals.

### 4.3 FACIAL EXPRESSION AND POSE SIGNIFICANTLY CONFOUND SOCIAL PERCEPTION

Unlike traditional correlational datasets such as FairFace and UTKFace, CausalFace enables the exploration of the influence of commonly thought of as confounding factors like lighting, pose, and facial expression.

Figure 3a illustrates the impact of these three variables on cosine similarities for various dimensions derived from the ABC and SCM models. The cosine similarity of a neutral text prompt is depicted by the grey dashed line, serving as a reference. Deviations from this reference, caused by adding a social perception item to text prompts, are represented as dots connected by thin grey lines. To support our interpretation of the visual insights drawn from Figure 3a, we employed linear regres-

sion models, accounting for a constant, to quantify the effects of confounding variables on social perception (Table B.1). As a face transitions from a grumpy expression to a smile, it is perceived in a more positive light and less negatively. In the leftmost panel of Figure 3a (Smiling), the slope is positive for positive-valence items ($\beta_+ = 1.03 \times 10^{-3}, p < 0.001$) and negative for their negative counterparts ($\beta_- = 1.03 \times 10^{-3} - 3.3 \times 10^{-4}, p < 0.001$). Lighting's influence on perception is less evident. The trajectories appear predominantly flat ($\beta = -1.1 \times 10^{-8}, p < 0.001$), suggesting that lighting might not significantly impact social perception. Regarding pose, a value of zero denotes a frontal stance, and negative/positive values correspond to left/right tilts. All dimension curves first slope negatively ($\beta_{<0} = -1.9 \times 10^{-3}, p < 0.001$) up to the midpoint, then symmetrically increase ($\beta_{<0} = 2.0 \times 10^{-3}, p < 0.001$). To interpret this effect, we need to consider that the actual cosine similarities are the highest for frontal poses given a neutral prompt and comparably low ($< 0.23$) for the most left and most right side poses (Figure B.2). In other words, for side-pose photos, the image embedding is more distant from the "person" mode in the embedding space. This makes the interpretation of changes induced by adding social adjectives to the text prompt challenging. However, we induce from this observation that pose variations are a significant source of noise that overlay other effects and could explain a significant share of the noise present in real-world datasets. Future research could address this observation in more detail.

As the effect of facial expression on social perception is most salient, we show an intersecting plot of race and gender and additionally vary facial expression (Figure 3b). Consistency is evident in the agency dimension's slope across intersecting groups, with women showcasing a more pronounced change as their smile varies. For "belief", only black males deviate from the pattern consistently observed for the remaining subgroups. As their smiles intensify, negative associations decrease. Notably, they are the only subgroup showing amplified positive associations with intensified smiles. The communion dimension's interpretation remains ambiguous.

## 4.4 VARIABILITY FROM PROTECTED ATTRIBUTES IS NOT GREATER THAN THAT FROM NON-PROTECTED ONES

CausalFace uniquely enables an analysis contrasting variations from legally protected attributes (e.g., age, race, and gender) with those typically seen as confounds. Figure 4 highlights the variation in cosine similarities (procedure described in Section 3.3) across these attributes, with the legally protected attributes showcased in orange and the non-protected attributes delineated in purple. While one might expect protected variables to induce stronger changes, tour analysis contradicts this intuition. To validate the inferences drawn from the visual representation, we conducted t-tests for all combinations of protected and non-protected variables (Table B.2).

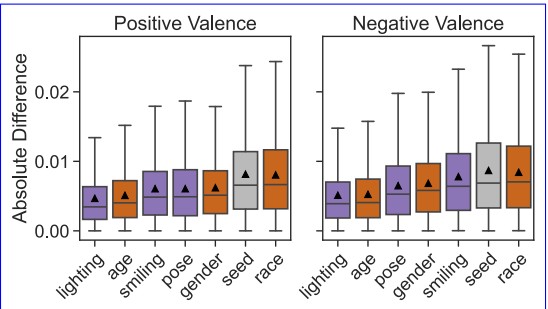

Figure 4: **Variation from protected attributes (orange) and non-protected ones (purple) are comparable in size.** Notably, age-induced variation is less pronounced than that from pose. See Figure B.3 for a more detailed analysis of single attributes in social perception models and for selected control attributes.

In sum, race causes the most pronounced differences throughout, while gender is overshadowed by pose and smiling ($p-\text{value} = 0.16$). Moreover, when focusing on negative valence, the variance attributed to age is comparable to that of lighting ($p-\text{value} = 0.25$) and produces the smallest variations among all examined variables.

Our observations suggest that non-protected attributes are important and should be taken into consideration to obtain a clear measurement of the socially relevant age- and gender-induced variations.

## 5 DISCUSSION

We causally investigated social perception bias in CLIP, anchoring our exploration in social psychology theories. We used three datasets: the real-image-based FairFace and UTKFace, and the

synthetically constructed CausalFace. FairFace and UTKFace are observational and allow computing correlations between protected attributes and social perception while not controlling for other attributes. By contrast, CausalFace allowed us to examine both legally protected characteristics and non-protected attributes, usually not studied by the bias research community, and to reach causal conclusions since attributes are controlled systematically and independently.

When investigating intersecting groups of age, race, and gender while controlling for other variables, CausalFace reveals distinct clusters, which suggest the presence of algorithmic bias. In contrast, these biases are not visible in FairFace or UTKFace, presumably because pose, lighting, and facial expression are not controlled there and introduce noise that obfuscates the signal.

Furthermore, we find that image-related confounding variables (i.e., lighting) have a comparably small stochastic impact on social perception. In contrast, subject-related non-protected variables, such as pose and facial expression, can have as strong of an impact on social perception as protected variables (age, gender, race). In particular, a shift from a grumpy to a neutral expression notably amplified positive associations while diminishing negative ones. Moreover, experiments altering the pose of a face show strong changes in CLIP's association of the image to a person in general. This effect potentially hinders the interpretation of social attribution and underlines the confounding effect of pose changes.

Our experiments reveal strong racial bias in CLIP, the variations attributed to gender are comparable to those of other non-protected attributes. Age introduces the least variations, in contrast with existing literature (Agarwal et al., 2021; Fraser et al., 2023), where it is considered a strong factor.

As a result, the discovery that biases related to legally protected attributes are not inherently more pronounced than those related to non-protected attributes conveys a crucial insight: For accurate measurement and protection of these attributes, it is essential that analyses rigorously account for all relevant confounding factors.

Generative models, in particular GANs and Diffusion Models, can now generate face images that look realistic to human observers, and one can effectively control some of the facial attributes. There are three caveats. First, manipulating one attribute may induce unwanted changes in other attributes; (Balakrishnan et al., 2021) show how to address this issue systematically, and it should be checked by visual inspection, as was done for the dataset we use (Liang et al., 2023). Second, it is possible that some physiognomies are generated more frequently than others. E.g., within the European group, it is possible that more Mediterranean-looking and fewer Scandinavian-looking physiognomies are generated. If this were a concern for a study, one would have to come up with a fine-grained definition of race and control this attribute directly, as our dataset does for the main racial groups. Third, it is possible that facial attribute manipulations are more or less effective for some or the other demographic groups. For the dataset we use, this was excluded by direct verification with human annotators (Liang et al., 2023, Fig. 3). Fourth, Meister et al. (2023) identify two primary confounding factors in gender research: pose and color. CausalFace directly addresses the pose, while it indirectly deals with color. Although CausalFace controls for image background, clothing, and hair color, thereby presumably minimizing color confounds, it does not explicitly eliminate them. Future research may directly tackle this aspect. Generative methods and techniques to validate their output are progressing fast. Thus, the experimental method we advocate, based on carefully controlled synthetic data, will become an increasingly attractive option for researchers.

Future research should extend our approach to a broader spectrum of vision-language models. Our study uses social perception as the primary outcome metric, revealing that smiling and pose significantly affect this measure. Research in human contexts shows that smiling, in particular, influences not only social but also job-related perceptions. Subsequent studies might explore practical metrics like occupation, examining how different levels of smiling and pose influence these factors in generative AI models. Additionally, synthetic datasets like CausalFace could enhance current debiasing methods (e.g., Berg et al., 2022) since they produce more precise measurements of bias. Future work could also address the design of synthetic datasets specifically for debiasing efforts, circumventing the cumbersome and costly annotation required for real-image datasets. These datasets should give researchers direct control over an increasing number of relevant variables, allowing a nuanced exploration of biases without additional costs.

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
