# A  IMPLEMENTATION DETAILS

## A.1  CLIP INFERENCE

All results were retrieved using CLIP ViT-B/32 as provided by OpenAI through the `openai-clip` Python library. The calculation of the cosine similarity of a set of images $I$ with a dimension $D$ is described in Equation (1). Table A.1 shows the full set of contextualized prompt templates used in this study. The dimensions sourced from theories of social perception used in this study are shown in Table A.2.

Table A.1: Prompt templates used for inferring text embeddings.

| |
| --- |
| A photo of a \<adjective\> person. |
| A \<adjective\> person. |
| This is a \<adjective\> person. |
| Cropped face photo of a \<adjective\> person. |

Table A.2: Adjectives per dimension. The Stereotype Content Model (SCM) and the ABC model both specify individual adjectives per dimension.

| SCM | | ABC | | | | | |
| --- | --- | --- | --- | --- | --- | --- | --- |
| Warmth | Competence | Agency + | Agency - | Belief + | Belief - | Communion + | Communion - |
| warm | competent | powerful | powerless | science-oriented | religious | trustworthy | untrustworthy |
| trustworthy | intelligent | high-status | low-status | alternative | conventional | sincere | dishonest |
| friendly | skilled | dominating | dominated | liberal | conservative | friendly | unfriendly |
| honest | efficient | wealthy | poor | modern | traditional | benevolent | threatening |
| likeable | assertive | confident | meek | | | likable | unpleasant |
| sincere | confident | competitive | passive | | | altruistic | egoistic |

## A.2  ADAPTED SC-WEAT

The traditional WEAT necessitates opposite pairs of target items. Although these pairs have been validated in human linguistics, their antonymic relationship in CLIP's embedding space remains unconfirmed. Given this uncertainty, we opted for the SC-WEAT, which obviates the need to define such pairs.

In the following, we draw upon Steed & Caliskan (2021) who adapts the WEAT (Caliskan et al., 2017) for investigating text-image cosine similarities. More specifically, we adapt the Single-Category WEAT (SC-WEAT) as follows: Let $A$ and $B$ be two sets of image embeddings of equal size (e.g., male and female faces), and let $x \in X$ be a single text embedding from a certain social perception category (e.g., warmth).

Then, we can calculate the test statistic for single text embedding as follows:

$$s(\boldsymbol{x}, \boldsymbol{A}, \boldsymbol{B}) = \text{mean}_{\boldsymbol{a} \in \boldsymbol{A}} \cos(\boldsymbol{x}, \boldsymbol{a}) - \text{mean}_{\boldsymbol{b} \in \boldsymbol{B}} \cos(\boldsymbol{x}, \boldsymbol{b}) \tag{2}$$

$$s(\boldsymbol{X}, \boldsymbol{A}, \boldsymbol{B}) = \text{mean}_{\boldsymbol{x} \in \boldsymbol{X}} s(\boldsymbol{x}, \boldsymbol{A}, \boldsymbol{B}) \tag{3}$$

The test statistic measures the differential association of the target $x$ concept with the attributes $A$ and $B$. The effect size (measured in Cohen's $d$) is:

$$es(\boldsymbol{x}, \boldsymbol{A}, \boldsymbol{B}) = \frac{s(\boldsymbol{x}, \boldsymbol{A}, \boldsymbol{B})}{\text{std}_{\boldsymbol{z} \in \boldsymbol{A} \cup \boldsymbol{B}} \cos(\boldsymbol{x}, \boldsymbol{z})} \tag{4}$$

$$es(\boldsymbol{X}, \boldsymbol{A}, \boldsymbol{B}) = \text{mean}_{\boldsymbol{x} \in \boldsymbol{X}} es(\boldsymbol{x}, \boldsymbol{A}, \boldsymbol{B}) \tag{5}$$

We test the significance of this association with a permutation test over all possible equal-size partitions $\{(\boldsymbol{A}_i, \boldsymbol{B}_i)\}_i$ of $\boldsymbol{A} \cup \boldsymbol{B}$ to generate a null hypothesis as if no biased associations existed. The one-sided $p$-value measures the unlikelihood of the null hypothesis:

$$p = \Pr[s(\boldsymbol{X}, \boldsymbol{A}_i, \boldsymbol{B}_i) > s(\boldsymbol{X}, \boldsymbol{A}, \boldsymbol{B})] \tag{6}$$

## A.3 MARKEDNESS

The concept of markedness was thoroughly reviewed by Wolfe & Caliskan (2022). How different groups of people form linguistic categories and what linguistic categories have utility varies widely between communities Bowker & Star (2000). This concept highlights how certain words and contexts in a language are deemed "unmarked" or more natural while others are "marked" as less natural or usual. In essence, unmarked elements carry meanings that are implicitly understood and need no explanation, whereas marked elements are considered unusual and require further clarification.

Markedness is quantified as the percentage preference for a neutral (unmarked) prompt over an attribute-specific (marked) prompt.

Consider an image featuring a person with a specific attribute, e.g. "white".

- Let $\boldsymbol{i}$ be the embedding of an image of a person with the given attribute.
- Let $\boldsymbol{t}_n$ be the embedding of an unmarked/neutral text prompt, e.g., "a photo of a person."
- Let $\boldsymbol{t}_m$ be the embedding of a marked text prompt, e.g., "a photo of a *white* person."
- Calculate the cosine similarity: $\cos(\boldsymbol{i}, \boldsymbol{t}_n)$ and $\cos(\boldsymbol{i}, \boldsymbol{t}_m)$.
- Repeat this process for all images in the dataset categorized under the given attribute.
- Count the number of times where $\cos(\boldsymbol{i}, \boldsymbol{t}_n) > \cos(\boldsymbol{i}, \boldsymbol{t}_m)$. Markedness describes the fraction of images where this condition is true.
- With $N$ being the total number of images of persons with the given attribute in the dataset, we can express the Markedness percentage $M\%$ as:

$$M\% = \frac{1}{N} \sum_{i=1}^{N} 1_{\{\cos(\boldsymbol{i}, \boldsymbol{t}_n) > \cos(\boldsymbol{i}, \boldsymbol{t}_m)\}} \times 100 \tag{7}$$

## A.4 SOCIAL PSYCHOLOGY LITERATURE ON STEREOTYPES

Social categorization and stereotyping are crucial in how we process information about people we encounter, allowing us to quickly infer attributes related to gender, race, and age Fiske & Neuberg (1990); Ito & Urland (2003). These inferences often activate prevalent stereotypes Bodenhausen et al. (2012); Macrae & Bodenhausen (2000), which are useful for navigating social interactions and decision-making Abele et al. (2021).

The Stereotype Content Model (SCM) posits Warmth and Competence as primary dimensions of our beliefs about stereotypes Fiske et al. (2002). Warmth assesses whether we see others as allies or foes, and Competence evaluates their ability to act on intentions. These dimensions are supported by various models, which provide a comprehensive understanding of stereotype content Abele et al. (2021); Koch et al. (2021).

Recent frameworks, such as the ABC model, propose Progressive-Traditional Beliefs as alternative central dimensions, with Warmth subdivided into Sociability and Morality, and Competence into Ability and Assertiveness Koch et al. (2016; 2020).

Demonstrably, different social groups thus are associated with distinct quadrants in the Warmth by Competence space, such that groups stereotyped as both high warmth and competence include the middle class, while those viewed as neither, include the homeless. Conversely, people with disabilities are seen as warm but not competent, and the wealthy as competent but not warm Fiske et al. (2002).

# B  ADDITIONAL RESULTS

## B.1  ALTERNATIVE BIAS METRICS

In the main paper, we presented the WEAT and markedness as bias metrics. Here, we present three additional bias metrics, following Geyik et al. (2019).

*Skew@k* measures the difference between the desired proportion of image attributes in $\tau_k^T$ and the actual proportion Geyik et al. (2019). For example, given the text query "this is a friendly person", a desired distribution of the image attribute gender could be 50%. Let the desired proportion of images with gender/race label $A$ in the ranked list be $p_{d,T,A} \in [0,1]$, and the actual proportion be $p_{\tau_T,T,A} \in [0,1]$. The resulting Skew of $\tau_T$ for an attribute label $A \in \mathcal{A}$ is

$$Skew_A@k(\tau_T) = \ln \frac{p_{\tau_T,T,A}}{p_{d,T,A}} \tag{8}$$

In other words, Skew@k is the (logarithmic) ratio of the proportion of images having the gender/race value $a_i$ among the top $k$ ranked results to the corresponding desired proportion for $A$. A negative Skew@k corresponds to a lesser than desired representation of images with gender/race $A$ in the top $k$ results, while a positive Skew@k corresponds to favoring such images.

*MaxSkew@k* describes the maximum skew among all attribute values Geyik et al. (2019).

The *NDKL* (Normalized Discounted Cumulative KL-Divergence) employs a ranking bias measure based on the Kullback-Leibler divergence, measuring how much one distribution differs from another (Kullback & Leibler, 1951). This measure is non-negative, with larger values indicating a greater divergence between the desired and actual distributions of attribute labels for a given $T$. It equals 0 in the ideal case of the two distributions being identical for each position. Let $D_{\tau_i}^T$ and $D^T$ denote the discrete distribution of image attributes race/gender in $\tau_i^T$ and the desired distribution, respectively. NDKL is defined by

$$NDKL(\tau_T) = \frac{1}{Z} \sum_{i=1}^{|\tau_y|} \frac{dKL(D_{\tau_{i_T}} \parallel D_T)}{\log_2(i+1)} \tag{9}$$

where

$$dKL(D_1 \parallel D_2) = \sum_j D_1(j) \ln \frac{D_1(j)}{D_2(j)} \tag{10}$$

is the KL-divergence of distribution $D_1$ with respect to distribution $D_2$, and $Z = \sum_{i=1}^{|\tau^r|} \frac{1}{\log_2(i+1)}$ is a normalization factor. The KL-divergence of the top-$k$ distribution and the desired distribution is a weighted average of *Skew at k* measurements (averaging over $A \in \mathcal{A}$).

The first row of Figure B.1 depicts Skew@k. Asians predominantly receive positive values, indicative of a favorable social perception relative to other races. The values for Blacks, however, are distinctly negative. In the case of Whites, CausalFace consistently shows negative social perception, while FairFace and UTKFace mostly hover around zero. In contrast, Skew@k for males and females hovers near zero, implying no significant bias towards either gender.

MaxSkew@k offers a slightly different perspective. Regarding race, CausalFace aligns more closely with FairFace, with UTKFace appearing as the outlier in this panel. A similar observation applies to gender.

Lastly, NDKL, akin to MaxSkew, reveals a more pronounced bias for race than for gender. The mean values across the datasets are closely aligned for gender but not for race. Unlike MaxSkew, however, the mean values for all three datasets appear similarly distant in the NDKL metric.

Overall, Skew@k, MaxSkew@k, and NDKL indicate that no dataset consistently emerges as an outlier. Only in one metric and for one race does CausalFace behave differently from the observational datasets. Therefore, we conclude that the three metrics reinforce the findings discussed in the main body of the paper: CausalFace does not significantly diverge from the observational datasets.

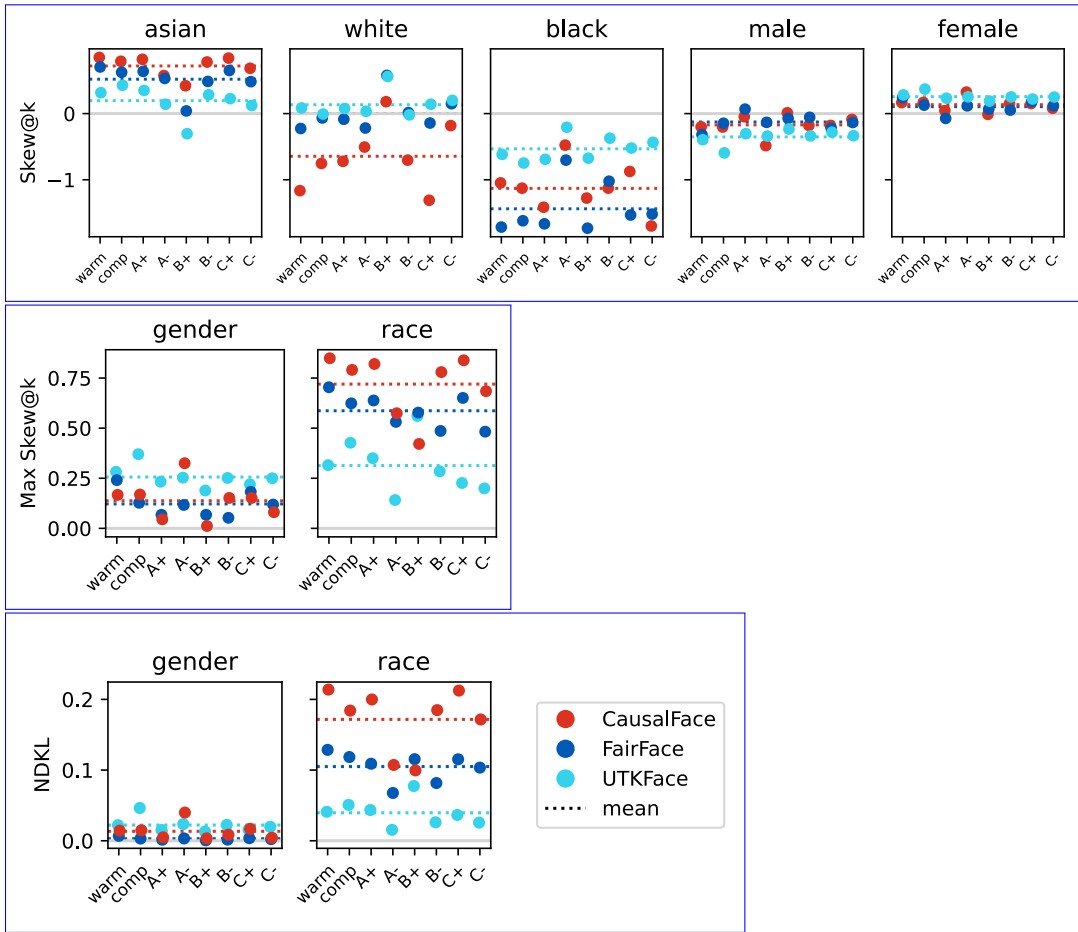

Figure B.1: A negative *Skew@k* corresponds to a lesser than desired representation of candidates with gender/race *A* in the top *k* results, while a positive Skew@k corresponds to favoring such images. We set k=1000. *NDKL* is a non-negative metric and equals 0 in the ideal case. Dotted lines represent the average values for each dataset.

## B.2    REGRESSION ANALYSIS OF CONFOUNDS

Figure 3a illustrates the influence of various confounds on social perception. To bolster the visual findings with statistical rigor, we perform multiple linear regressions. Each regression model includes a constant term, formulated as: social perception $\sim \beta_0 + \beta_1 \times$ confound. For smiling, we conduct separate regressions for positive and negative valence. Further, we split the data based on whether the smiling values are above or below 0. In the cases of pose and lighting, we do not differentiate between valences since the trends appear consistent across both. However, for pose, we again stratify the analysis based on values being above or below 0.

Concerning smiling, interestingly, when expressions transition from a neutral to a cheerful state, social perception of both positive and negative valence dimensions tends to diminish. However, transitioning from a grumpy to a neutral expression witnesses an increase in the positive valence perception and a concurrent decrease in the negative valence perception—a trend that mirrors intuitive human interpretations. In terms of lighting, a negative slope in social perception is observed when transitioning from dim lighting to more focused illuminations on specific facial regions. Lastly, regarding pose, social perceptions are found to rise when moving from a left tilt (less than 0) to a frontal stance (equal to 0) but show a reduction as the pose transitions from this frontal position to a right tilt (greater than 0).

Table B.1: Regression analysis assessing the impact of confounds on social perception. For smiling, separate models are estimated for positive $(+)$ and negative $(-)$ valence items. For pose, separate models are conducted for left-tilted faces $(< 0)$ and right-tilted faces $(> 0)$, with $0$ representing a frontal view.

|  |  | coef | std err | $P > |t|$ | $R^2$ |
|---|---|---|---|---|---|
| smile | const | -1.6e-02 | 3.2e-04 | 4.3e-21 | 7.1e-01 |
|  | $\beta_+$ | **1.0e-03** | 1.6e-04 | 3.4e-06 | 7.1e-01 |
|  | const | -1.5e-02 | 9.0e-05 | 1.5e-15 | 8.7e-01 |
|  | $\beta_-$ | **-3.3e-04** | 4.4e-05 | 7.5e-05 | 8.7e-01 |
| lighting | const | -1.6e-02 | 6.4e-05 | 0.0e+00 | 1.5e-03 |
|  | $\beta_l$ | **-1.1e-08** | 7.2e-10 | 6.9e-51 | 1.5e-03 |
| pose | const | -1.4e-02 | 8.2e-05 | 0.0e+00 | 3.5e-02 |
|  | $\beta_{<0}$ | **-1.9e-03** | 4.2e-05 | 0.0e+00 | 3.5e-02 |
|  | const | -1.4e-02 | 8.1e-05 | 0.0e+00 | 3.9e-02 |
|  | $\beta_{<0}$ | **2.0e-03** | 4.2e-05 | 0.0e+00 | 3.9e-02 |

## B.3 RAW IMPACT OF CONFOUNDS ON SOCIAL PERCEPTION

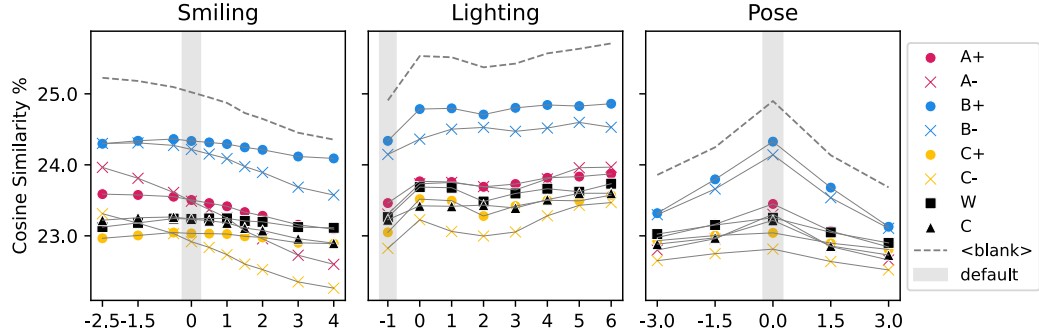

Figure B.2: Variation of Figure 3a without subtracting $< blank >$ baseline values. The dashed grey line depicts the average cosine similarities to prompts describing a person without specifying any dimension adjective ("blank"). The grey vertical box denotes the default value. Smiling shows strong differential effects on positive and negative valence. Lighting shows a moderate positive trend. Pose shows that CLIP attributes the image the most to a person if it is shown in a frontal view.

## B.4 COMPARISON BETWEEN VARIATION INDUCED BY CONFOUNDS AND LEGALLY PROTECTED ATTRIBUTES

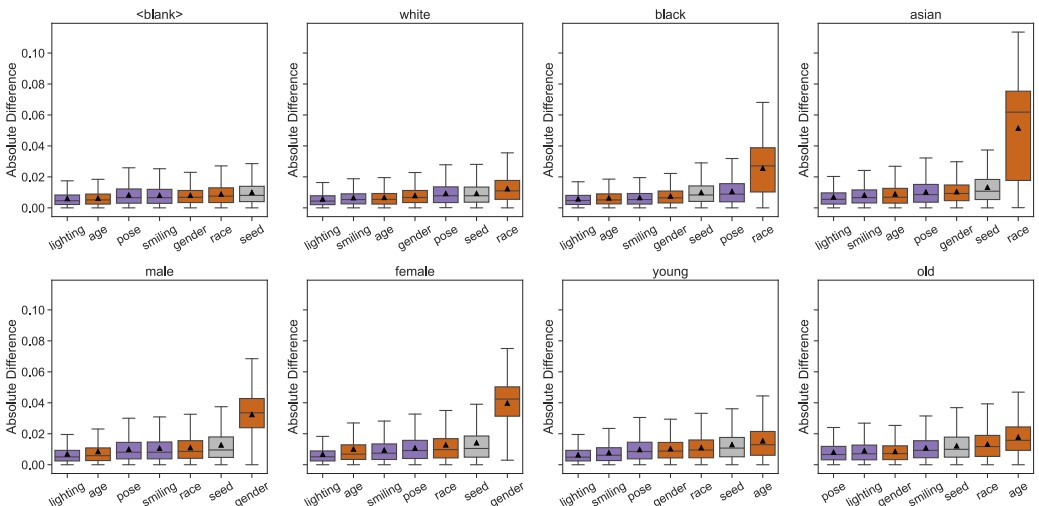

(a) Selected control attributes. Titles depict the prompt with the first subplot representing a neutral prompt, while the second subplot requests a photo of a white person.

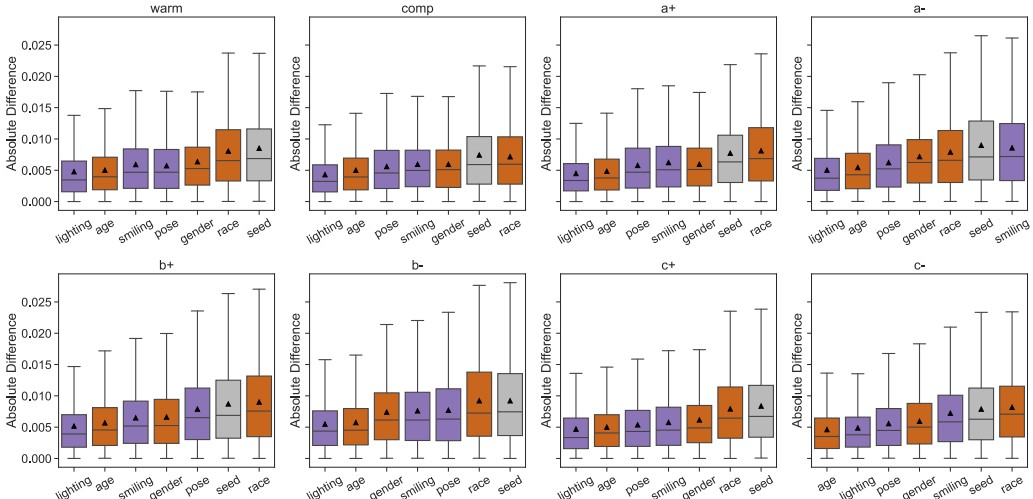

(b) Dimension of social perception sourced from SCM and ABC models, indicated by the titles of each subplot.

Figure B.3: More detailed impact of attribute changes (extended version of Figure 4).

## C ON THE CORRELATION OF POSITIVE AND NEGATIVE VALENCE CATEGORIES

### C.1 EMBEDDING STRUCTURE OF DIMENSION ADJECTIVES SOURCED FROM THEORETICAL MODELS

In our study, the main bias metric is constructed by calculating cosine similarities between images and adjectives. Those adjectives are sourced from two theoretical frameworks: ABC and SCM. SCM outlines 'warmth' and 'competence' as the key dimensions to quantify stereotypes in human cognition. Each dimension is described by six specific adjectives—for example, *warmth* by 'friendly,' 'warm,' and 'likable'; *competence* by 'intelligent,' 'skilled,' and 'assertive.' Alternatively,

Table B.2: Comparison of differences in cosine similarities: Non-protected vs. protected attributes. Results derived from pairwise two-sided t-tests, with only non-significant ($p$−value $> 0.05$) differences, showcased.

| Valence | Non-protected | Protected | t-statistic | $p$-value |
|---|---|---|---|---|
| Positive | pose | gender | -1.39 | 0.16 |
| | smiling | gender | -1.41 | 0.16 |
| Negative | pose | gender | -1.39 | 0.16 |
| | smiling | gender | -1.41 | 0.16 |
| | lighting | age | -1.15 | 0.25 |

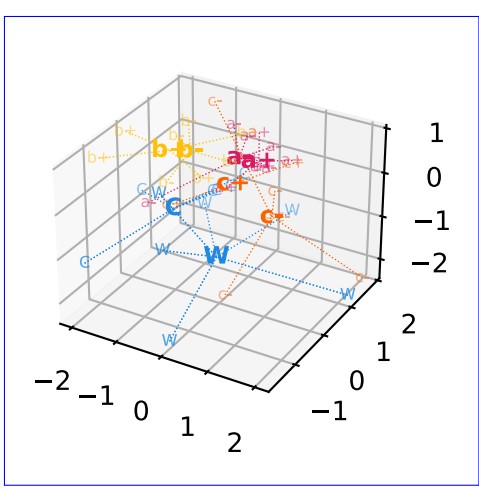 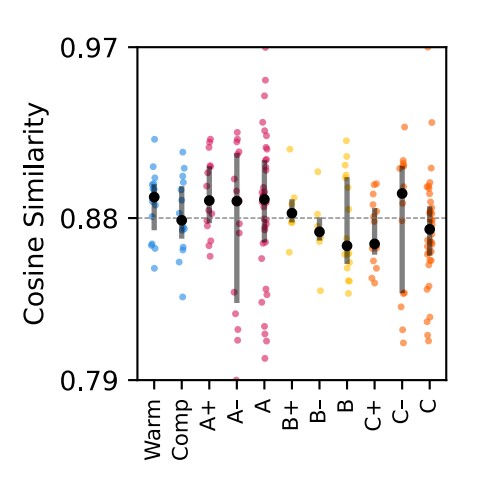

Figure C.4: Left Panel: 3D visualization of text embeddings across theoretical models. Each marker denotes an adjective positioned by its PCA-reduced embedding. The center for each dimension, calculated as the mean of its items, is connected to its respective adjectives by dashed lines. Colors indicate the average cosine similarity for each dimension's items. Right Panel: Cosine similarities of text-to-text pairs. Each colored dot indicates the cossine similarity of one text-to-text pair. Within-valence similarities are indicated by the x-tick markers A+, A-, B+, B-, C+, C-. Cross-valence similarities are indicated by markers A, B, C. Furthermore, the black dots indicate averages within dimension, the grey vertical segments indicate interquartile ranges, and the grey horizontal line indicates the median of all dimensions.

Table C.3: $p$-values from a one-sided t-test comparing within-valence cosine similarities (positive-negative) to cross-valence similarities. $p$-values $> 0.05$ suggest that cross-valence similarities are not significantly smaller than within-valence, indicating proximity of positive and negative valence items in the embedding space.

| Category | pos-cross | neg-cross |
|---|---|---|
| Agency | 0.29 | 0.80 |
| Belief | 0.23 | 0.59 |
| Communion | 0.56 | 0.31 |

ABC introduces the three dimensions of 'Agency,' 'Belief,' and 'Communion,' and additionally distinguishes within dimension between positive valence adjectives like 'powerful' and 'wealthy' and negative ones such as 'poor' and 'meek.' The comprehensive adjective list and their corresponding dimensions across both frameworks are detailed in Table A.2.

Before using those adjectives to quantify bias in images, we want to understand how the dimensions are reflected within CLIP's embedding space. To do this, we investigate within and cross-valence

similarities. Specifically, let $d_1^+$ and $d_2^+$ be two text embeddings of adjectives of the positive valence agency dimension, and $d_1^-$ and $d_2^-$ be text embeddings of adjectives of the negative valence agency dimension. The within-valence similarity is given by calculating the cosine similarities $\cos(d_i^+, d_j^+)$ for all positive adjective pairs $(i, j)$. The cross-valence similarity is determined by calculating $\cos(d_i^+, d_j^-)$ for all mixed adjective pairs $(i, j)$.

Intuitively, we would assume that within-valence cosine similarities are higher than cross-valence similarities. However, $t$-tests comparing pairs of within- to cross-valence similarities indicate that cross-valence similarities are never significantly lower than their within-valence counterparts (Table C.3). In other words, positive and negative valences are close in space.

A 3D visualization of text embeddings (Figure C.4) supports these findings: positive and negative valences are closely clustered. In combination with CLIP's modality gap between text and image embeddings (Desai et al., 2023), we can expect that cosine similarities between an image and a positive and negative adjective are positively correlated.

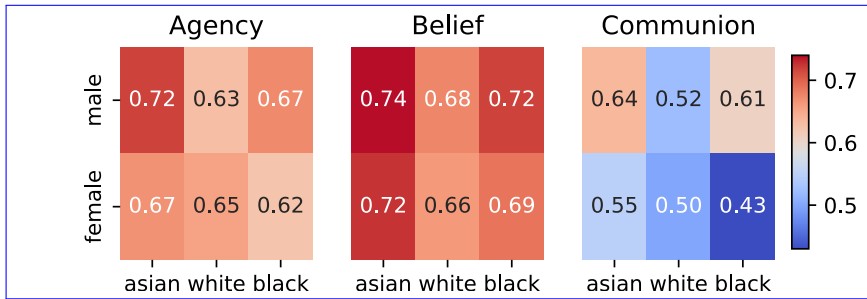

Figure C.5: Heatmap of Pearson correlation coefficients of positive and negative valence dimensions of the ABC model.

Furthermore, we are interested in how these high correlations differ across dimensions and gender-race groups. To that end, we compute the Pearson correlation coefficients for each intersecting gender-race group across the three dimensions of agency belief and communion. Figure C.5 presents these coefficients in a heatmap. Two key observations emerge from this analysis. First, within each subplot, the highest magnitude is noted in the upper-left corner, corresponding to Asian males, while lower correlations are observed in the column representing white individuals. Across all heatmaps, a consistent pattern is evident: the dimension of belief demonstrates the strongest correlation, whereas communion exhibits the weakest correlation between positive and negative valence adjectives. This latter finding aligns with expectations, given that communion adjectives are more akin to political dichotomies, like conservative versus liberal, rather than linguistic opposites.

## D    DATASET VISUALIZATIONS

### D.1    FAIRFACE AND UTKFACE

Out of the races present in Fairface, we used the original "White" and "Black" categories and aggregated "East Asian" and "Southeastern Asian" into a single category. "Indian", "Hispanic" and "Middle Eastern" were not used in this study. Similarly, we use "Asian", "White", and "Black" but not the "Indian" and "Others" categories. For both datasets, we only used those images that are annotated with age 20 or older. This is motivated by matching the general demographic distribution available in CausalFace.

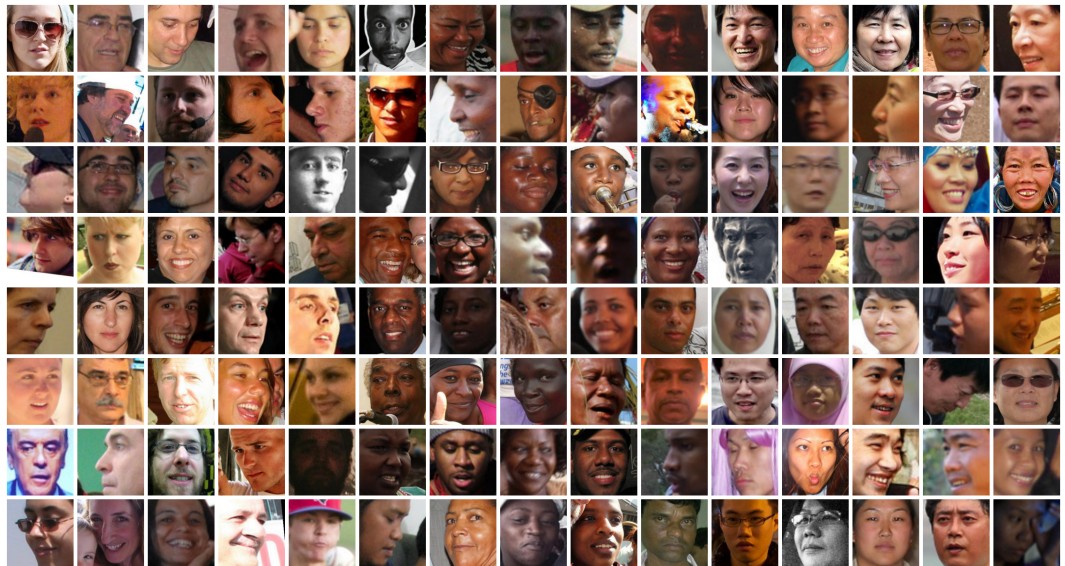

Figure D.1: Fairface examples.

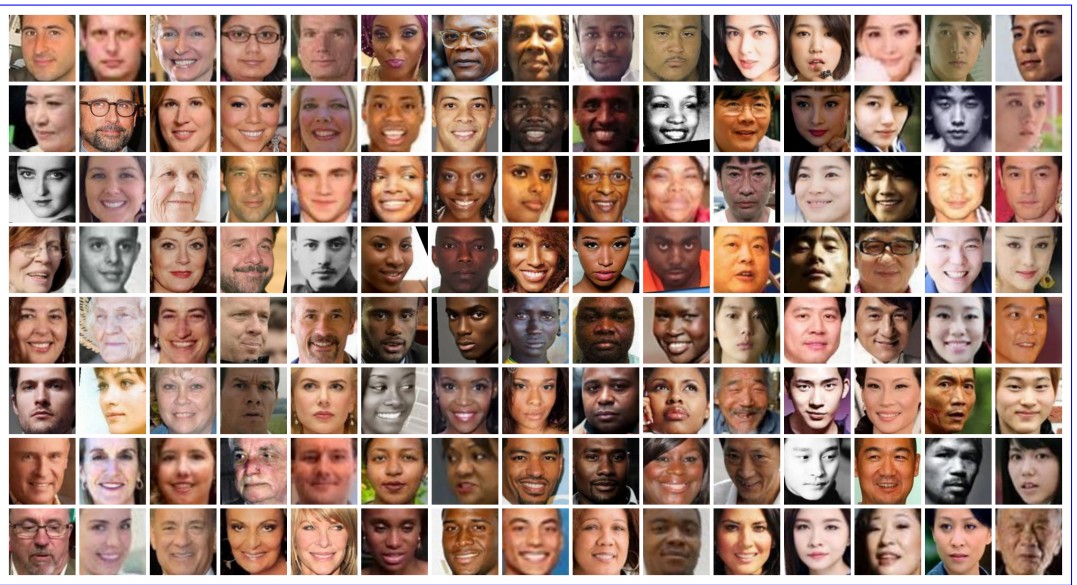

Figure D.2: UTKFace examples.

## D.2 CAUSALFACE

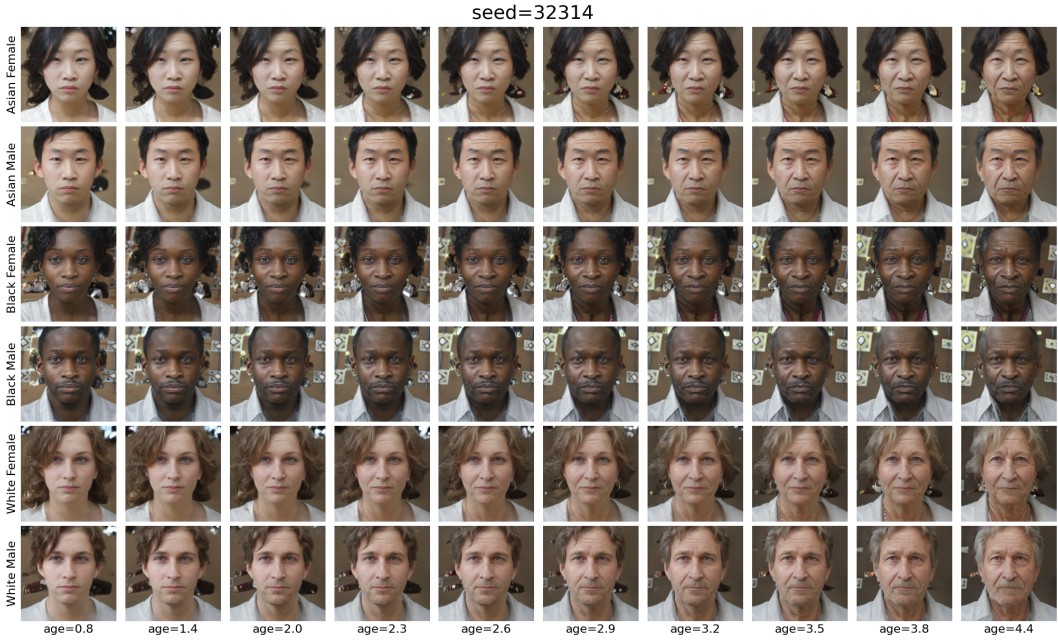

Figure D.3: CausalFace example images along the "age" dimension.

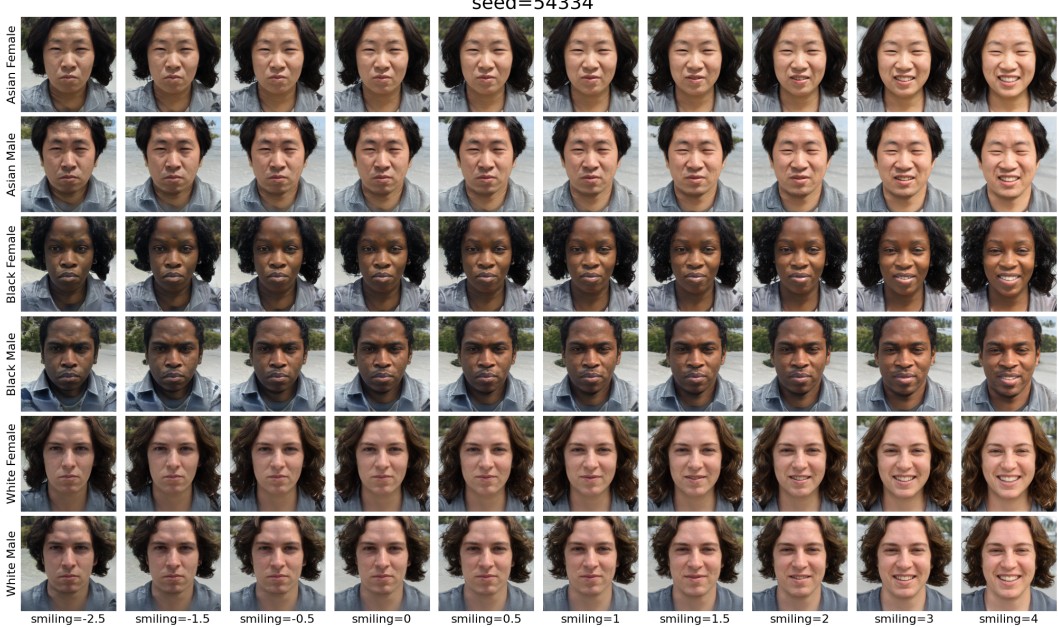

Figure D.4: CausalFace example images along the "smiling" dimension.

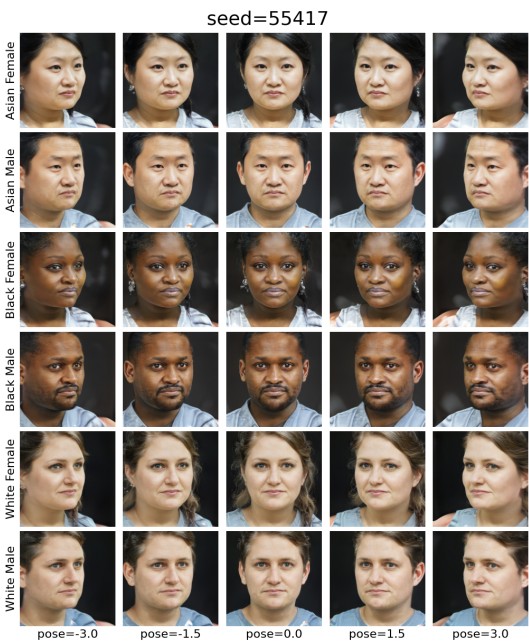

Figure D.5: CausalFace example images along the "pose" dimension.

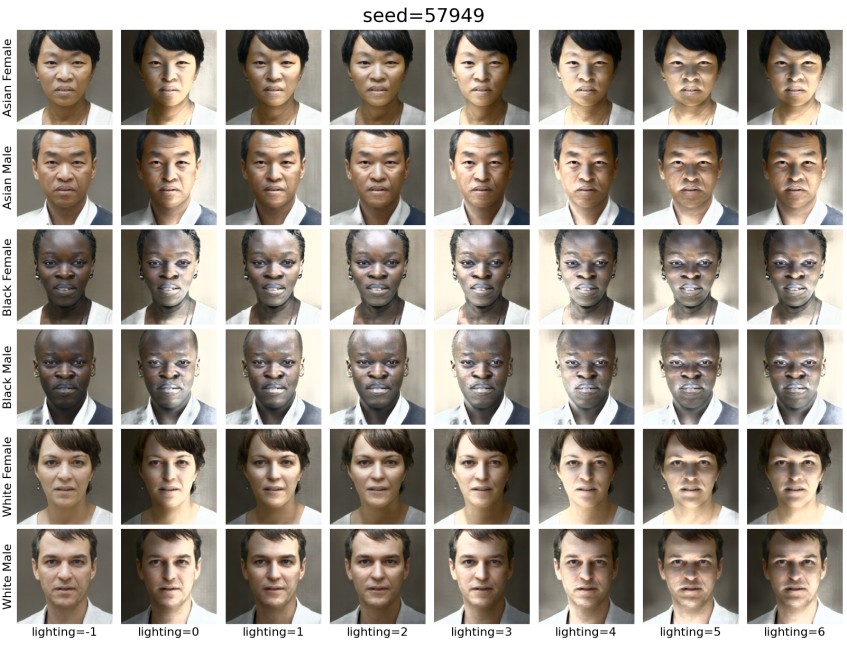

Figure D.6: CausalFace example images along the "lighting" dimension.