# OpenReview forum: "Causal analysis of social bias in CLIP"
_ICLR.cc/2024/Conference — Submitted to ICLR 2024_

### Official Review · Reviewer_mYoV · 2023-10-24

**Soundness:** 2 fair
**Presentation:** 3 good
**Contribution:** 1 poor
**Rating:** 3
**Confidence:** 3

**Summary:**

The paper analyzes social bias in CLIP image and text embeddings by comparing the cosine similarity between images of faces and attributes extracted from social psychology. The paper claims to study causality rather than correlations using a synthetic dataset, CausalFace.

**Strengths:**

The paper is well-written and well-motivated, it is easy to read and understand and it does a good job at positioning itself with respect to the related work, which is relevant and up-to-date.

**Weaknesses:**

W1. The novelty of the paper may be limited, as there is already abundant literature studying correlations between image and attribute embeddings in CLIP (many already cited in the related work). The contributions of this study are 1) using attributes from social psychology instead of a self-defined list of words, and 2) claiming causality instead of correlation. The first contribution may not be enough by itself, whereas the second contribution is challenged in the following point.

W2. The paper argues that the analysis conducted on the CausalFace dataset, in contrast to FairFace, implies causality and not only correlation because the elements/confounders in CausalFace can be controlled by the generation process. However, it could be argued there is not enough evidence to assume there are no confounders in CausalFace just because it has been generated synthetically.

Specifically, images in CausalFace have been generated using a GAN by imitating a training distribution. The training distribution may very likely present biases, which could be learned and transferred to the synthetic images. Hence, concluding that using synthetically generated images implies causality may not stand. It would be interesting to discuss how the potential biases in the image generation algorithm can affect the results of this study.

W3. I would say the main paper is not self-contained. Many necessary details to understand the flow of the paper and its conclusions are placed in the supplementary material. This is a subtle way of evading the 9-page limitation. The supplementary material should be used for extra information only. Some examples:
- The metrics computation, especially the cosine similarity between the image and attribute embeddings is essential to understand the results in, e.g. Table 1, but its definition is in Appendix A1.
- The whole section 4.1 discusses results that are not in the paper but in Appendix B.

**Questions:**

- I am curious to know why the authors chose to use the phrase “*legally* protected attributes” to refer to demographic attributes such as race, gender, or age, as the adjective *legal* (i.e. referring to the law) has different interpretations in different places of the world.

- Why only use 3 groups (Asian, Black, White) from FairFace instead of all the data?

---

> ### Author Response · Authors · 2023-11-16
> **Initial Reply Addressing Identified Weaknesses and Questions.**
>
> We thank reviewer mYoV for their insightful remarks.
>
> **W1**: The reviewer's observation that our paper employs ``attributes from social psychology instead of a self-defined list of words" may not fully acknowledge the depth and scientific rigor associated with using psychometrically validated scales; We refer to [Furr (2021)](https://books.google.ch/books?hl=en&lr=&id=xto9EAAAQBAJ&oi=fnd&pg=PT14&dq=Psychometrics:+An+Introduction&ots=FeGG0pXykH&sig=qeu28wVVW03Ji9G1Hzv3H3_t2SE&redir_esc=y#v=onepage&q=Psychometrics%3A%20An%20Introduction&f=false) for an introduction to the field of Psychometrics. Moreover, the widespread adoption of the SCM and ABC models underscore their practical relevance in measuring stereotype perception. Thus, our decision to utilize validated scales in our study is not a mere preference for a different list of words but a deliberate choice to enhance the contribution of our research to the field.
>
> **W2**: Yes, it is important to check for biases, artifacts, and confounds. If the referee has specific concerns, we will be the first ones to want to hear about them. The reviewer asks for "evidence of absence of confounders". Unfortunately, that is typically impossible to come by. However, the evidence for the presence of confounders is precious. We selected the Liang (2023) dataset precisely because it was carefully controlled. They verified that attribute manipulations were equally effective across different demographics using the ratings of five annotators per image (Fig. 3, Liang 2023). They checked for visible image artifacts (Sec 4.1) and excluded all images containing artifacts from their dataset. We have added to the manuscript a sentence discussing this (please also see response to referee 3zN2). Are we sure that all possible confounds and biases have been eliminated? Of course not. That would be foolish. In the history of science, there are many examples of important experiments where confounds and biases were hammered out year after year by generations of scientists. The crucial point is adopting the scientific method, which is based on experimentation, and experiments may be refined over time. We argue that measurements of bias in AI models now can, and thus must, move beyond correlations. Generative methods offer a practical way forward toward controlled experiments and causal conclusions. Thanks to progress in generative AI and constructive scrutiny from an engaged community of scholars, we expect our measurements to become more accurate with time.
>
> **W3**: We have thoroughly revised our paper in light of the reviewer's critique. We concur on the need to clarify the definition of cosine similarity, therefore we introduced equation (1) in section 3.1. Although our paper also utilizes metrics like WEAT, mean cosine similarities, and markedness, we have chosen to keep their detailed descriptions in the appendix. This decision aligns with common academic practices of detailing mathematical methodologies in supplementary materials, rather than in the main text, especially since these metrics are well-established and not novel contributions of our work. Section 4.1 was moved to the appendix, now appendix C, as we think it is not part of the main results of our paper but rather a supporting analysis that helps interpreting the results in Fig. 2.
>
> **Q1**: Despite variations in national laws, global human rights standards set by the United Nations—like UDHR, ICCPR, ICESCR, and CEDAW—mandate non-discrimination based on race, age, and gender. While some countries may not legally protect these attributes, the majority do. Hence, we refer to these variables as legally protected, aligning with the scholarly consensus in various fields.
>
> **Q2**: Our study compares FairFace to CausalFace; The latter however only includes three races: Asian, White, and Black. Some researchers might be interested in analyzing social perceptions across all FairFace races without comparing them to CausalFace. We will open-source our code, which allows those researchers to adapt our experiments to all racial categories in FairFace.

---

> ### Comment · Reviewer_qwQ3 · 2023-11-20
> **Raises a good point re: confounders**
>
> Just wanted to chime in here that I think reviewer mYoV raises a good point about the possibility of confounders in this generated dataset making this signal less clean. I think this is actually fairly essential to this paper and something I hadn't thought of in my review. The authors response that this is something considered in the original dataset is good and I appreciate that it was added to the discussion section - in my opinion, given how important this is, the evidence that these confounders are minimal should be moved up to 3.2 where the CausalFace dataset is introduced.
>
> Extra thought: the authors are quite dismissive of this point in their rebuttal and the response around the "history of science" comes off as somewhat condescending imo. In fact, this is an extremely reasonable and important critique of the paper on which the legitimacy of its contribution rests, and it was something which was not addressed at all in the original draft. In the "history of science", I agree that experiments are not guaranteed to come without confounds, but this mostly applies to observed phenomena (e.g. natural experiments): here, we are dealing with entirely controlled settings and should be aiming for a higher bar, particularly given the (strong) claims of the paper.

---

> > ### Author Response · Authors · 2023-11-21
> > **Thank you and follow up**
> >
> > Thank you for sharing your thoughts on this topic. We agree that this is a crucial aspect of our research and acknowledge that our initial draft did not sufficiently highlight the rigorous validation processes established by Liang et al. (2023). We appreciate your suggestion to incorporate more detail in Section 3.2 and have added the following:
> >
> > "The dataset's unique feature is its ability to alter image attributes independently, facilitating causal interpretations. To validate this, Liang et al. (2023) visually confirmed that modifying one attribute does not inadvertently alter others. Moreover, human annotators verified the consistency of facial attribute modifications across different demographic groups (Liang et al., 2023, Fig. 3). A more detailed discussion on the dataset's causal interpretation is available in the discussion section."
> >
> > We will soon upload an updated draft reflecting these changes.
> >
> > Our interpretation of reviewer mYoV's comments is that we need to provide proof that there are no confounds, which is, unfortunately, impossible. We regret if our previous response seemed condescending. We were concerned that this important methodological point would be missed. We assure all reviewers that their comments are appreciated and well-received. To address any ongoing concerns, we offer additional clarification below:
> >
> > Synthetic datasets inherently lack a guarantee against confounds, a topic widely discussed in the literature. However, methods exist to address confounds when identified, as illustrated by Balakrishnan et al. (ECCV 2020, [https://arxiv.org/pdf/2007.06570.pdf)]). Their work demonstrates techniques to eliminate spurious correlations, such as those between hair length and gender (Fig. 5) and other examples (Figs 18 and 19; hair length and beard in men, or gender and earrings).
> > Diligent qualitative inspection and systematic annotation of synthetic data can help identify and address confounds and correlates, a process Liang et al. (2023) have effectively undertaken.
> >
> > We hope that we can convince the reviewers that the existing approach, i.e., correlational analysis on wild-collected and annotated data, has significant weaknesses: It is very difficult to stamp out confounds and spurious correlations and intersectional studies are often impossible (E.g., it is difficult to collect a large enough dataset containing the necessary diversity of viewpoints, facial expressions, ages for all the combinations of protected attributes).
> >
> > Therefore, we are convinced that our observations are already safer and cleaner than those one can reach with traditional observational datasets. Furthermore, the method we propose (experimental, based on synthetic datasets) is bound to improve with time as techniques for image synthesis improve. This is why we are keen to propose our study to the ICLR audience.
> >
> > We are eager to communicate effectively with our readers and welcome further questions, challenges, and comments that will aid in clarifying our findings.

---

> ### Comment · Reviewer_mYoV · 2023-11-22
> **Response to authors**
>
> The reviewer thanks the authors for their response.  The reviewer acknowledges the time and effort invested in elucidating the raised concerns. While the reviewer values the comprehensive insights provided in the response, the reviewer has concerns about the choice of tone in response W2. The reviewer understands that discussions in the scientific community merit a respectful and collaborative tone. The reviewer's intentions were only to raise constructive criticism to improve the scientific contribution of the reviewed work.
>
> In the following, the reviewer addresses the specific points raised in the response:
>
> W1: The initial review did not intend to overlook the significance and scientific rigor linked with the utilization of psychometrically validated scales. On the contrary, the original review explicitly recognized this aspect as a primary contribution of the paper. The central query posed in the initial review pertains to whether this particular contribution, in isolation, constitutes a sufficient basis to warrant submission to ICLR. Regrettably, the authors have not yet addressed this specific concern in their response.
>
> W2: The reviewer remains unconvinced regarding the assumption that synthetic datasets do not introduce artifacts or confounds. This concern is fundamental, given the paper's assertion of studying causality rather than correlation. It is incumbent upon the authors to provide a robust justification to assure readers that the study indeed warrants claims of causality. Notably, ample evidence suggests the presence of gender artifacts in visual datasets across various image levels [1]. Consequently, it remains unclear whether the utilization of a synthetic dataset eliminates these artifacts or if they are learned and transferred within the generative model.
>
> W3: Thanks for reviewing the paper to make it self-contained.
>
> Q1 and Q2: Thank you for answering the questions.
>
> Reference: [1] Meister et al. Gender Artifacts in Visual Datasets. ICCV 2023.

---

> > ### Author Response · Authors · 2023-11-22
> > **Thank you; adding further explanation to W1**
> >
> > We appreciate the time and effort reviewer mYoV has invested in critically reviewing our paper and responses. Our mutual goal is clear: we aim to maintain a respectful and collaborative tone in our discourse. Ultimately, we aim to contribute a paper that propels the community forward. We think the discussion and review process is essential to this endeavor. The insightful concerns and challenges raised by mYoV have enhanced our paper.
> > We regret any perception that our tone was inappropriate and wish to reassure you that our comments were made in the spirit of constructive collaboration.
> >
> >
> > **W1:** We thank the reviewer for providing details on W1. We didn't presume that reviewers would be familiar with psychometric validation, so it's encouraging to find that the reviewer is well-versed in these areas. We are thankful for the reviewer's clarification. Our methodology includes template averaging, a standard practice to query CLIP (e.g., [Berg et al. 2022](https://arxiv.org/abs/2203.11933)). We use multiple sentence templates like "A photo of a warm person" or "A picture of a warm person" and then average these to mitigate noise and enhance stability.
> >
> > We take this approach a step further by suggesting averaging qualitative words to solidify outcome stability further. This prompts an essential question: How should we select the attributes to be averaged? We suggest leveraging theories on social perception. These theories have successfully quantified key human cognition dimensions like warmth and competence, providing validated attributes (e.g., friendliness and likability for warmth) to detail these dimensions.
> >
> > Concurring with the reviewer, we recognize research that explores the correlations between attribute items and image embeddings. Our contribution to this body of work is twofold: firstly, we advocate for using psychometrically validated items; secondly, we propose averaging these items to achieve a robust and stable representation of each dimension. We think that the ICLR community will find value in both these contributions.
> >
> > May the reviewer specify why they ask us to assess this contribution "in isolation"? We believe that a paper's value exceeds the sum of its individual components. In our case, the synergy between theoretical foundations and experimental methodology is integral and forms—as a whole—the basis of our submission to ICLR.

---

> > > ### Author Response · Authors · 2023-11-22
> > > **Thank you; adding further explanation to W2**
> > >
> > > **W2:** We sincerely appreciate the reviewer's frank and constructive feedback, highlighting their ongoing reservations. This enables us to articulate our arguments better.
> > >
> > > In their first answer, the reviewer states, "there is not enough evidence to assume there are no confounders in CausalFace." Similarly, in their second response, the reviewer writes that they are "unconvinced [...] that synthetic datasets do not introduce artifacts."
> > > Addressing these concerns, we wish to clarify (again) that our study does not claim the absence of confounders in CausalFace. We state explicitly the opposite. Quoting from our initial reply to reviewer mYoV: "Are we sure that all possible confounds and biases have been eliminated? Of course not." Quoting from our November 21 reply: "Synthetic datasets inherently lack a guarantee against confounds."
> > > Furthermore, in the following, we give examples of confounds found by Balakrishnan et al. (2020) in their synthetic dataset. We hope this leaves no doubt in the reviewer's mind on our position. Is there anything in our manuscript that may induce confusion? We will be grateful to the reviewer for help identifying sentences/paragraphs of our paper that may lead to the impression that we were asserting "no confounders." We would greatly appreciate any suggestions for rephrasing.
> > >
> > > What is true of synthetic datasets is that one may hunt for confounds and spurious correlations and develop methods to fix such problems, as Balakrishnan et al. (2020) did (please see their Fig 5). In our first response to the reviewer, we explain that Liang et al. (2023) looked carefully for correlations and confounds by qualitative inspection and using painstaking crowdsourced annotations. They eliminated any problems they found in their dataset. Their paper was published in ICCV2023 and went through a rigorous refereeing process.
> > > The reviewer has not commented specifically on these measures undertaken and we would be greatly interested in their thoughts.
> > >
> > > We agree completely with the reviewer that in-the-wild datasets have been amply documented as containing spurious correlations and biases (We thank the reviewer for the Meister et al. (2023) citation--we cite it now in our introduction and discussion.) Additionally, in-the-wild datasets do not allow intersectional studies, as we explained in our follow-up from Nov. 21.
> > >
> > > Please allow us to elaborate on causality: The experimental method, where one variable at a time is manipulated, is the only way to establish causality in the general case (see, e.g., Pearl's book). Generative methods allow us to manipulate one variable at a time, and thus carry out experiments, and thus establish causal links between an attribute of the image and the behavior of the algorithm. By contrast, in-the-wild datasets do not give us that level of control and only allow us to compute correlations.
> > > In response to the reviewer's question: The value of our paper is to introduce in this literature a first example of an experimental approach. We hope that our example will inspire others.
> > >
> > > In the latter part of their feedback, the reviewer identifies a specific type of artifact, transitioning the discussion from the presence of any artifacts to the presence of specific ones. We are grateful for this precise articulation of concerns, as it allows us to address them directly.
> > > In response to the artifact of average pose highlighted by Meister et al. (2023), our study has taken deliberate measures to address this issue. By intentionally controlling for pose as a non-sensitive attribute, we effectively eliminate it as a confounding factor in our synthetic dataset. Figure D.4 in our paper visually confirms where identical poses for gender variations are clearly demonstrated. We include Meister et al. (2023) as a reference, as it reinforces the claims of our paper.
> > >
> > > Moving to the average color gender artifacts mentioned by Meister et al. (2023): CausalFace aligns background, clothing, and hair and skin color, which we believe significantly reduces potential color confounds. Similarly to the point concerning pose, the claims mentioned by Meister et al. (2023) strengthen the proposition of using an artificially generated dataset over one collected in the wild. Nonetheless, this potential confound is not one that Liang et al. (2023) explored and it deserves attention. Had this point been raised earlier in the rebuttal process, we would have had the opportunity to address it. As it stands, the best we can do is raise this point in the discussion and point future research towards it.

---

### Official Review · Reviewer_qwQ3 · 2023-10-31

**Soundness:** 4 excellent
**Presentation:** 3 good
**Contribution:** 4 excellent
**Rating:** 8
**Confidence:** 4

**Summary:**

This paper conducts an analysis of bias in CLIP using a generated dataset of faces, leveraging its generated nature to draw more precise, causal-type conclusions. They examine cosine similarity in CLIP space to various words from well-known social psychological frameworks. They present a number of empirical takeaways, including that conclusions from finely-controlled generated data can differ from pre-existing, real data and that several confounding variables (e.g. "smiling") can also affect bias measurements.

**Strengths:**

- this is an interesting empirical analysis which makes some important points about bias evaluation
- makes intelligent choices in evaluation design, a good example of what this can look like
- interesting takeaways around difference between synthetic and real data, as well as confounding variables

**Weaknesses:**

- I think the title should be a little more precise - specifically this is around "social bias in CLIP for face images" or something, it's not studying all areas of bias
- Figure B.1 was unclear to me - I'm not sure how to read the point about within vs cross valence similarities off this plot
- Sec 4.2: I get a little confused about a few of the evaluation procedures here. For instance, I think "mean cosine similarities" could be explained a bit more - I can guess what it might be but don't know for sure. Also I think markedness is not explained so clearly: not clear what "relative preference frequency" is or a "neutral prompt" - again I could guess but should be written out.
- I think the point about positive correlation between positive and negative terms is a good one, I think this could be shown more strongly with a "control group" of words: is it that some images are just correlated to all words? words of a certain type?
- the word "intensity" is used without definition - what does this mean?



Small notes:
- typos: bottom of p1: 'sd',
- Sec 3.4: unclear how these thresholds are determined or what they really mean (e.g. 0.7 for age)
- what is the x-axis in Fig 3a?
- Fig 3b and commentary at end of 4.4: if I'm reading this correctly, most groups negative associations decrease as smiles intensify - it's stated here that this is unusual for black males.

**Questions:**

- I think the observation that "the widely held belief that age- and gender-induced variations are strong factors needs to be reconsidered" is misguided (end of 4.5 and 5). The authors seem to believe that social factors like gender receive special attention since they are causes of particularly strong deviations - I think this misses an important point. It isn't that these factors necessarily cause the largest deviations (of course other visibly salient factors will matter), it's that these factors are important for socially determined reasons, and therefore measuring model impacts and mitigating disparities/harms where they exist matters more

---

> ### Author Response · Authors · 2023-11-16
> **Initial Reply Addressing Identified Weaknesses, Small Notes, and Question.**
>
> We thank reviewer qwQ3 for their positive assessment and insightful remarks.
>
> **W1**: You are inviting us to make our title more specific and we agree:  *"A causal analysis of social bias towards faces in CLIP."*
>
> **W2**: We agree with the reviewer that the figure caption (now Figure C.3)  was unclear. The interpretation of the figure was moved to the appendix and a mathematical definition was added: *"Specifically, let $a^+_1$ and $a^+_2$ be two text embeddings of attributes of the positive valence agency dimension, and $a^-_1$ and $a^-_2$ be text embeddings of attributes of the negative valence agency dimension. The within-valence similarity is given by calculating the cosine similarities $ \cos{(a^+_i, a^+_j)}$ for all positive attribute pairs \( (i,j) \). The cross-valence similarity is determined by calculating $ \cos{(a^+_i, a^-_j)}$ for all mixed attribute pairs \( (i,j) \)."*
>
> **W3a**: We agree that mean cosine similarities were not defined sufficiently. For clarification, we moved Eq. 1 from the appendix to the main paper (see. Sec. 3.1.). In addition, we added further explanation in the results section when discussing Table 1, which reads as follows: *"Mean cosine similarities are calculated as the mean of cosine similarities between an image category and all attribute categories of SCM and the positive attribute categories of ABC (see Equation (1))"*.
>
> **W3b**: We added appendix A.3 to further explain and mathematically define markedness.
>
> **W4**: We show the positive correlation of positive and negative valence attributes in Figure 2. The high correlation is however only visually assessed. We agree with the reviewer that we should calculate a correlation coefficient. We did so for each dimension (agency, belief, communion) and for each intersecting group separately. We plot and interpret a heatmap of correlations in Figure C.4.
>
> **W5**: Indeed, it was not obvious what the term "intensity" means in the context or study. Therefore, we added the following explanation when it was first introduced: *"In this context, we use valence as the direction of change in cosine similarity (increasing or decreasing) and intensity as the magnitude or absolute difference in change."*
>
> **SN2**: To maintain consistency with Liang (2023), we kept the original scales for smiling (-2.5 to 4) and age (0.8 to 4.4), as shown in Figures D.2 and D.3. For smiling, the non-equidistant levels range from -2.5 to 4. We selected a threshold of 0.7 to distinguish between sufficiently different smiling faces, deliberately excluding the 0.5 increments. A similar approach was applied to age. To avoid confusing a future reader, we introduced the following footnote: *"The effect of the thresholds can be better understood by looking at the scale of Figures D.2 and D.3"*
>
> **SN3**: To avoid redundancy between the title and the x-axis label in Figure 3a, the x-axes are unlabeled. To clarify, we have added to the figure caption: *'The three panels depict the variations in smiling, lighting, and pose on the x-axis.'*
>
> **SN4**: We agree that we should have been more precise with the use of language at this point. The patterns of all gender-race intersections look similar, with one exception: Male-blacks. Only for this group does smiling lead to an increase in positive social perception. The interpretation needs to be more nuanced. There is a subtle but existing difference between stating a "glass is half full" versus "a glass is half empty". Similarly, there is a subtle but existing difference between describing someone with less negative terms or more positive terms. Black males are the only group that is described predominantly with more positive terms as the smile increases. We changed the corresponding description in the paper to: *"For "belief", only black males deviate from the pattern consistently observed for the remaining subgroups."*
>
> **Q1**: We appreciate the reviewer's insightful feedback on our discussion of variations in legally protected versus non-protected variables. We acknowledge that the original phrasing might have led to misunderstandings. We concur with the reviewer that gender, race, and age are socially significant factors, meriting the focused research they receive. Their importance indeed underscores our call for greater precision in measurement and a thorough examination of potential confounds in statistical analyses. To clarify our position, we have revised the statement at the end of section 4.4: *"Our observations suggest that non-protected attributes are important and should be taken into consideration to obtain a clear measurement of the socially relevant age- and gender-induced variations."*

---

> > ### Comment · Reviewer_qwQ3 · 2023-11-20
> > **Response**
> >
> > Thanks for this rebuttal - I appreciate the clarifications and the paper (still) looks good to me. I'm going to follow up on a discussion in another review that I think raises a good point.

---

### Official Review · Reviewer_3zN2 · 2023-11-01

**Soundness:** 3 good
**Presentation:** 2 fair
**Contribution:** 3 good
**Rating:** 6
**Confidence:** 4

**Summary:**

The paper studies biases in CLIP using a synthetic dataset of images generated by GAN, which they refer to as CausalFace. The authors argue that synthetic data offers the opportunity to control for possible confounding factors, such as lighting and pose, leading to a more accurate assessment of biases in CLIP. In their experiments, they consider different labels (e.g. "friendly", "conservative", etc) that are borrowed from prior works in social psychology (namely the the ABC Model and the Stereotype Content Model) and compare their embeddings with the image embeddings using cosine similarity.

Using this setup, the authors present several interesting findings. First, they find that non-protected attributes, such as pose and smiling, can have a non-trivial impact. In particular, the variance in the cosine similarity by changing lighting, smiling, and pose is comparable to the variance induced by changing protected attributes. In fact, the impact of "pose" is stronger than the impact of "age." The authors study the effect of such factors at a greater depth; e.g.  image-related confounding factors, such as lightning, have less effect that subject-related factors, such as pose and smile. So, non-protected attribute can cause a significant amount of noise when doing bias-related analysis of CLIP. Second, the authors show that because such confounding factors can be controlled in CausalFace, new patterns emerge in CausalFace that are not visible in datasets like FairFace. This is demonstrated, for example, in Figure 2.

Overall, the paper offers an interesting and useful insight when studying biases in multimodal systems. It highlights that confounding factors need to be taken into account, and that correlational studies would typically underestimate biases in models such as CLIP (because of the noise introduced by the confounding factors).

**Strengths:**

The paper offers several interesting insights in a topic that is becoming increasingly important. The experimental results are convincing, and the overall message is quite useful to the community. The authors take care in handling several potential issues; e.g. by demonstrating that synthetic data are statistically similar to real images, among others.

**Weaknesses:**

- The first limitation is that the authors use a single dataset only in their analysis, which is FairFace. There are other datasets such as UTK Face and CelebA that can be included to support the argument further. It's not clear if the conclusions in FairFace would continue to hold so showing that they hold in other datasets would strengthen the argument. In addition, MIAP (https://paperswithcode.com/dataset/miap) is quite different from FairFace in that it collects images in natural settings (not just face images) so I would expect those confounding factors to be even more prominent in that dataset. This would be a useful message to point out.
- There are important places that need further clarity. For instance, can you please provide a precise mathematical definition of "markedness," similar to how WEAT is defined in the appendix? Page 5 explains it a bit but a precise definition should be included. Also, which specific prompts did the authors use to create various levels of "pose," "lightning", and "smile"? They are not described in the paper or the appendix as far as I can see.
- The authors focus on sentiment related attributes, such as warmth, communion, competence, etc. It would make sense for these attributes to be impacted by confounders, such as pose and smiling. But, there are other equally important association biases, such as relating gender with occupation, that may not be as sensitive to those issues. These are not studied in the paper.

**Questions:**

- What does "neutral lightning" mean? It seems from Figure 3 that neutral lightning may lie at the middle of the scale, which would reveal a pattern that is different from random noise.
- When comparing the impact of "smiling" in Figure 3.b, why did you choose to compare -1.5 with 3.0 instead of going for the extreme ends; i.e. -2.5 with 4.0?
- In Section 4.4,  the authors claim that "black males" deviate from the expected pattern in Figure 3.b because they are "the only subgroup showing amplified positive associations with intensified smiles." I'm curious to know why the authors think this is unexpected? Shouldn't the fact that the other groups don't exhibit this be the unexpected pattern?
- Which specific prompts did the authors use to create various levels of "pose," "lightning", and "smile"?

---

> ### Author Response · Authors · 2023-11-16
> **Initial Reply Addressing Identified Weaknesses.**
>
> We thank reviewer 3zN2 for their positive assessment and insightful remarks.
>
> **W1**: We appreciate the reviewer's suggestion regarding the selection of face datasets for our analysis. In response, we have thoroughly evaluated each of the proposed datasets and would like to share our rationale behind their consideration: MIAP does not have face crops and, therefore, would introduce a significant amount of noise from contextual elements. These are additional confounding factors that could result in visible differences compared to FairFace. It is not the main goal of our paper to control for these types of confounds; instead, we focus on face-related confounds. MIAP would, therefore, not directly support our findings, and thus, we decided to focus on face-cropped datasets. CelebA meets our criteria but we do not consider it ideal for assessing biases in race, gender, or age groups for the following reason: It is likely that celebrities occur in the training data of CLIP. Therefore, we would assume that embeddings are overly specific to individuals rather than representative of broader demographic groups. Unlike FairFace, which includes only a limited selection of celebrities, CelebA is comprised entirely of celebrity individuals. We recognize UTKFaces as a viable alternative to FairFace for our analysis. We are in the process of extending our study to include this dataset and plan to integrate these results before the end of the rebuttal period.
>
> **W2a**: Thank you for highlighting the need for clarity in the definition of markedness. To address this, we have added Appendix A.3 which succinctly reviews related works, illustrates the concept with examples, and provides a precise mathematical definition.
>
> **W2b**: We do not generate synthetic images ourselves but the dataset was pre-generated and made available to us by Liang (2023).
> Smiling, lighting, and pose are dimensions generated through latent space traversals using a GAN to create face images. The specific manifestations of these dimensions in the images are illustrated in Appendix D. Upon reviewing our paper's section on image datasets, we acknowledge a potential source of confusion. Initially, we mentioned "we generate variations," which we have now revised to *"we sample variations"* to better reflect our methodology.
>
> **W3**: The reviewer hypothesizes that cosine similarities between warmth/competence-related text prompts and images might fluctuate more significantly due to varying smile/pose levels than those between job type-related prompts and images with similarly varying smiling/pose levels. This is an interesting hypothesis and we do not investigate the relation of smiling/pose to occupation in generative AI and we are not aware of literature that does so. However, research investigating humans suggests that facial attributes do systematically vary with occupation: Facial appearance matters for leader selection [(Stocker 2016)](https://journals.plos.org/plosone/article?id=10.1371/journal.pone.0159950). More specifically, the width-to-height ratio seems to distinguish CEOs [(Lewis 2013)](https://www.sciencedirect.com/science/article/abs/pii/S0191886912000049) .
> Furthermore, men and women smiled more in response to a low-status job than to a high-status job [(Vrugt 2002)](https://onlinelibrary.wiley.com/doi/10.1002/ejsp.99)--suggesting that smiling may be a more significant predictor of job type than gender. More generally, smiling has been found to differ across age and gender [(Morse 2010](https://www.tandfonline.com/doi/abs/10.1080/00223980.1982.9915318),
> [LaFrance 2003](https://pubmed.ncbi.nlm.nih.gov/12696842/),[Otta 1998](https://pubmed.ncbi.nlm.nih.gov/9923165/),
> [Doff 1999](https://link.springer.com/article/10.1007/BF03395325),
> [DeSantis 2005)](https://pubmed.ncbi.nlm.nih.gov/16342596/). Therefore, we assume that occupation is also sensitive to smiling/pose.
> Consequently, there is no clear answer to the question of which outcome metric is more sensitive to varying levels of smiling/pose.
> This is an interesting question to ask therefore, we add the following paragraph to the discussion: *"Our study uses social perception as the primary outcome metric, revealing that smiling and pose significantly affect this measure. Research in human contexts shows that smiling in particular, influences not only social but also job-related perceptions. Future research could focus on practical metrics such as occupation, investigating how varying degrees of smiling and pose impact these aspects in generative AI."*

---

> > ### Author Response · Authors · 2023-11-16
> > **Initial Reply Addressing Identified Questions.**
> >
> > **Q1**: We could not locate the exact use of 'neutral lighting' in our text, could you please point it out to us? Figure C.5 illustrates varying lighting levels, with the leftmost column (-1) representing the 'default lighting' that is also used in all other images that are not modified along the lighting dimensions. Lighting values 0-6 describe specific lighting from different directions. We welcome the reviewer's insight on how these variations might introduce systematic variation rather than random noise, as we have yet to find a fitting interpretation.
> >
> > **Q2**: We have corrected the error in the legend. Upon review, it's evident that the plot includes all smiling levels from -2.5 to 4.0, as can be confirmed by counting the markers of various sizes.
> >
> > **Q3**: We agree that we should have been more precise with the use of language at this point. The patterns of all gender-race intersections look similar, with one exception: Male-blacks. Only for this group does smiling lead to an increase in positive social perception. The interpretation needs to be more nuanced. There is a subtle but existing difference between stating a "glass is half full" versus "a glass is half empty". Similarly, there is a subtle but existing difference between describing someone with less negative terms or more positive terms. Black males are the only group that is described predominantly with more positive terms as the smile increases. We changed the corresponding description in the paper to: *"For ``belief'', only black males deviate from the pattern consistently observed for the remaining subgroups."*
> >
> > **Q4**: We refer to our answer to W2b.

---

> ### Comment · Reviewer_3zN2 · 2023-11-20
> **Thanks**
>
> Thanks for the reply. Adding UTKFace would strengthen the findings of the paper. Also, thanks for clarifying the misunderstanding about generating the samples, and fixing it in the paper. I agree that adding the last paragraph to the discussions section would be useful to the reader.
>
> By "neutral lighting," I meant a value of -1 in Figure 3(a), which you have now clarified.

---

> > ### Author Response · Authors · 2023-11-22
> > **Followup on W1: We included UTKFace**
> >
> > We again thank the reviewer for pointing us to the inclusion of UTKFace. We have completed our revision of the manuscript.
> >
> > **W1**: Analyzing UTKFace prompted us to add columns to Table 1 and additional (light blue colored) dots to Figures 1 and B1, as well as an additional row in Figure 3. The addition of UTKFace to the bias metrics is intriguing. It supports the findings that CausalFace is not structurally different from those two observational datasets. Regarding Figure 2, similar to FairFace, UTKFace shows no consistent ordering with respect to age.
> >
> > We believe that the inclusion of an additional dataset significantly enhances the robustness of our paper, and we are hopeful that the reviewer will share this perspective.

---

### Official Review · Reviewer_SWyU · 2023-11-03

**Soundness:** 3 good
**Presentation:** 3 good
**Contribution:** 3 good
**Rating:** 6
**Confidence:** 2

**Summary:**

In this work the authors investigate social perception biases in CLIP through the lens of social psychology. For this analysis, the authors use real-images and synthetic image datasets. The analysis reveal several interesting findings: pose and facial expression can have strong social perception, legally protected variables do not introduce greater biases than non-protected attributes.

**Strengths:**

**1. Originality:** I found this paper sufficiently novel. It deviates from the traditional way of measuring biases and systematically controlling the factors while measuring the biases.

**2. Well explained motivation:** The motivation provided in the introduction is excellent. It lays out the  disadvantages of the current bias studies and how they aim to solve those.

**3. Strong analysis along multiple axes:** The authors did an excellent job in providing detailed analysis along multiple axes eg: intersectional biases in Figure-2.

**Weaknesses:**

**1. Clarity:** I found the paper little difficult to follow especially in the sections 3.2 in which the authors describe about theoretical frameworks of social phycology. The authors can provide more context about these frameworks and how are they related in the proposed study in the form of examples, figures etc.

**2. Unaddressed questions about biases in synthetic datasets:** The authors didn't include discussion about potential limitations in the synthetic faces created using GANs.

**Questions:**

1. Can these findings be applicable to other vision-language models?

2. Is there any reason to authors not using bias metrics like max-skewness or NDKL?

---

> ### Author Response · Authors · 2023-11-16
> **Initial Reply Addressing Identified Weaknesses and Raised Questions**
>
> We thank reviewer SWyU for their positive assessment and insightful remarks.
>
> **W1**: To address the reviewer's concerns concerning clarity, we have made the following revisions: Firstly, we revised section (formerly) 3.2/ now 3.1 and additionally include mathematical notation. Additionally, we improved the caption of Figure B1, which now conveys our methodology more transparently. Secondly, recognizing the constraints of paper length, we have introduced Appendix A.4, which contains a detailed discussion of social psychology's view on stereotypes, and add additional references. This should provide an in-depth background for readers with a specific interest in the theoretical underpinnings of our study.  We also made concise changes to section 2.2 to logically connected both models by stating:
> *"A more recent framework, the ABC Model, Koch(2016), proposes beliefs as alternative central dimension, and subdivides the Warmth and Competence differently into two categories."* Furthermore, we introduce both theories by stating that *"social psychology has long been recognized for delineating the primary dimensions of beliefs about stereotypes."* Thirdly, we justified the integration of social psychology frameworks within our study as follows in 2.2: *"Given the reflection of human-like biases in Generative AI, our study employs two leading theoretical frameworks from social psychology—long recognized for quantifying cognitive biases—to measure stereotypes systematically"*. In all sections, we have striven for consistency in terminology, using "dimensions" and "attributes" to describe the components of the SCM and ABC models consistently. We hope these changes address your concerns, and we invite your further feedback to refine our manuscript.
>
> **W2**: Thank you for pointing this out. Yes, we should have commented on this. We have included the following statement in the discussion: *"Generative models, in particular GANs and Diffusion Models, can now generate face images that look realistic to human observers, and one can effectively control some of the facial attributes. There are three caveats. First, manipulating one attribute may induce unwanted changes in other attributes;  Balakrishnan (2021) show how to address this issue systematically, and it should be checked by visual inspection, as was done for the dataset we use (Liang 2023). Second, it is possible that some physiognomies are generated more frequently than others. E.g., within the European group, it is possible that more Mediterranean-looking and fewer Scandinavian-looking physiognomies are generated. If this were a concern for a study, one would have to come up with a fine-grained definition of race and control this attribute directly, as our dataset does for the main racial groups. Third, it is possible that facial attribute manipulations are more or less effective for some or the other demographic groups. For the dataset we use, this was excluded by direct verification with human annotators (Liang 2023, Fig. 3.). Generative methods and techniques to validate their output are progressing fast. Thus, the experimental method we advocate, based on carefully controlled synthetic data, will become an increasingly attractive option for researchers."*
>
> **Q1**: One could expect that bias stems from training data (i.e., text-image pairs) rather than the model architecture. Therefore, we would expect that our findings apply to other models if they are trained on similar datasets. We present our method primarily as a tool for investigating biases in such models. Following the reviewer's question, we have updated the discussion section to emphasize the applicability of our approach to future research on a broader range of vision-language models.
>
> **Q2**: Besides WEAT, we considered max-skewness and NDKL as alternative bias metrics but eventually omitted these results. Responding to the reviewer's interest, we are now reintegrating these findings into the paper and will present them before the rebuttal period concludes.

---

> > ### Author Response · Authors · 2023-11-22
> > **Follow up on Q2**
> >
> > **Q2**: We welcomed the question proposed by reviewer SWyU on including additional bias metrics. The bias metrics proposed by [Geyik et al. (2019)](https://arxiv.org/abs/1905.01989), namely Skew@k, MaxSkew@k, and NDKL, provide conclusions similar to the initially proposed ones. We have moved the explanation and evaluation of these metrics to Appendix B1.
> > We believe that including additional bias metrics significantly enhances the robustness of our paper, and we are hopeful that the reviewer will share this perspective.

---

### Author Response · Authors · 2023-11-16
**TLDR; Brief Summary of Most Important Changes.**

We thank all reviewers for their valuable feedback. We replied to each reviewer's comments and questions individually. In addition, we uploaded a revised version of our manuscript with the following changes with the main goal of improving clarity:

* The title was slightly modified to be more precise as the original one was a bit too broad, as noted by reviewer qwQ3.
* We moved Equation 1 from the supplementary material to the main text in order to make our methodology clearer to the reader.
* The analysis of the positive correlation of attributes with opposing valence was moved from the main paper to the supplementary material. We do not see them as the main finding of our paper but rather as a supporting analysis to interpret other results.
* We provided a more precise definition of ``markedness" and a more elaborative summary of social psychology literature on stereotypes in the supplementary material.
* Several minor improvements to improve clarity. Changes are highlighted in blue in the manuscript.

---

> ### Author Response · Authors · 2023-11-22
> **Final Summary of Most Important Changes.**
>
> We wish to express our sincere thanks to all four reviewers. We believe that our paper has significantly improved, and we hope the reviewers share this perception.
>
> We uploaded a final revision of our manuscript. Below, we summarize the main changes between the current and final revision and the initial version of the paper:
>
> 1) We introduced three additional bias metrics, defined them mathematically, added a new figure in Section B1, and interpreted the results, which align with the initially chosen bias metrics.
>
> 2) We evaluated a second observational dataset, namely UTKFace, leading to changes in Table 1, Figures 1 and 2. The interpretation of this dataset is in line with the interpretation of FairFace.
>
> 3) We have substantially expanded the discussion section of the paper, which now includes a discussion of the evaluation procedure and the causal interpretation of the dataset. It also addresses several points raised by the reviewers that are not covered in the main paper and should be pursued in future research.
>
> 4) The interpretation of Figures C4 and C5 were moved to the appendix. Equation (1) was shifted from the appendix to the main paper.
>
> 5) The entire paper was revisited with striving for consistency in the terms describing social perception dimensions and their comprising items. The word "attribute" is now exclusively used for legally protected (like race and gender) or non-protected attributes.
>
> 6) We have added a section on the correlation between positive and negative valence dimensions to the Supplementary Materials (Appendix C), including a newly introduced Figure C5.
>
> 7) We have included a detailed definition of Markedness in Section A3 and a comprehensive review of social psychology literature in Section A4.

---

### Public Comment · ~Manuel_Knott1 · 2024-08-27
**Preprint available**

Readers who are interested in the latest version of this work can find the current preprint on arXiv:
http://arxiv.org/abs/2408.14435

---

### Meta-Review · Area_Chair_vWYE · 2023-12-21

**Metareview:**

The main issues include the paper's limited novelty, as it primarily extends existing literature without offering substantial new insights. Reviewers also expressed concerns about the paper's assumption that synthetic datasets like CausalFace inherently eliminate confounders, which may not be a valid claim. Furthermore, important details and definitions are relegated to supplementary material, making the main paper less self-contained. AC carefully reviewed the discussion between reviewer mYoV and authors, and found that the concerns were not addressed. Also, AC feels that the analysis per se is not "causal" but still statistical.

Considering its average score is below the acceptance margin, AC recommends reject.

**Justification For Why Not Higher Score:**

mainly due to its invalid confounder construction and non-causal analytical methodology.

**Justification For Why Not Lower Score:**

N/A

---

### Decision · Program_Chairs · 2024-01-16

Reject